**Brief Communication**

# Open-top multisample dual-view light-sheet microscope for live imaging of large multicellular systems

Franziska Moos[1,2,4], Simon Suppinger [1,2,4], Gustavo de Medeiros [1,3], Koen Cornelius Oost[1], Andrea Boni[3], Camille Rémy[3], Sera Lotte Weevers [1,2], Charisios Tsiairis[1,2], Petr Strnad [3] ✉ & Prisca Liberali [1,2] ✉

Multicellular systems grow over the course of weeks from single cells to tissues or even full organisms, making live imaging challenging. To bridge spatiotemporal scales, we present an open-top dual-view and dual-illumination light-sheet microscope dedicated to live imaging of large specimens at single-cell resolution. The configuration of objectives together with a customizable multiwell mounting system combines dual view with high-throughput multiposition imaging. We use this microscope to image a wide variety of samples and highlight its capabilities to gain quantitative single-cell information in large specimens such as mature intestinal organoids and gastruloids.

Visualizing single-cell dynamics shaping complex tissues and understanding the underlying mechanisms is an overarching goal in biology. However, these complex biological phenomena often cross large spatiotemporal scales, as multicellular systems can grow over the course of days. Furthermore, biological processes and especially in vitro models are often affected by sample-to-sample heterogeneity. A microscope for live imaging of such systems must provide high throughput within each experiment to draw robust conclusions. Additionally, it must provide sufficient spatiotemporal resolution and image quality for large light-scattering samples while minimizing light dosage and keeping the sample accessible. Light-sheet microscopy overcomes some of these challenges due to its low phototoxicity and high optical sectioning[1–3]. For large specimens, multiview or SimView light-sheet microscopy has provided improved image quality by acquiring images from opposing directions using sample rotation or multiple objective lenses[4–7]. These techniques are, however, limited in throughput[8]. Further, open-top[9], inverted[10–12] or single-objective approaches such as oblique plane[13], SCAPE[14] or DaXi[15] light-sheet microscopes have been developed to enable multisample imaging, in which the sample is supported from the bottom[10,11] while granting direct accessibility from the top. However, these systems do not allow imaging from opposing detection sides.

Here we present an open-top, dual-view and dual-illumination light-sheet microscope, combining the advantages of multiview imaging with an open-top geometry and a multiwell sample holder enabling long-term multiposition three-dimensional (3D) live imaging of large specimens. We show its capabilities to achieve high image quality in a variety of model systems such as intestinal, liver and salivary gland organoids, gastruloids, *Hydra* and human colon cancer organoids, reaching sizes of up to 550 μm and recordings for up to 12 days. We obtain quantitative features and present a detailed single-cell analysis through tracking and segmentation for intestinal organoids and gastruloids.

This microscope contains two opposing illumination objectives (Nikon 10×, numerical aperture (NA) 0.2, effective NA 0.06) each tilted slightly upward from the horizontal plane, illuminating the sample from two sides, and two opposing detection objectives imaging from two directions (Nikon 16×, NA 0.8: the system is also mechanically compatible with Nikon 25×, NA 1.1) (Fig. 1a–e and Extended Data Figs. 1 and 2). This geometry creates space above the illumination objectives (Fig. 1b,e) for a sample holder containing an array of up to four sample chambers (Fig. 1f). Immersion medium (water) is placed in a reservoir filling the space between the detection objectives. To obtain two opposing light-sheets illuminating the sample at the largest possible

[1]Friedrich Miescher Institute for Biomedical Research, Basel, Switzerland. [2]University of Basel, Basel, Switzerland. [3]Viventis Microscopy Sàrl, Lausanne, Switzerland. [4]These authors contributed equally: Franziska Moos, Simon Suppinger. ✉e-mail: petr.strnad@viventis-microscopy.com; prisca.liberali@fmi.ch

angle minimizing striping artifacts and illuminating uniformly, we used super-long working distance air objectives, coupled the illumination light into the immersion medium through a glass window and designed a correction triplet lens compensating for aberrations. The objective area has environmental control (humidity, temperature and $CO_2$). An additional beam path uses one of the detection objectives as a condenser illuminating the sample to acquire transmitted light images. With this objective configuration, the resolution is limited by sampling (measured lateral full-width at half-maximum (FWHM) 0.8 μm; Extended Data Fig. 3). Technical specifications are listed in Supplementary Table 1. For sample mounting, we developed customizable chambers produced from fluoroethylene propylene (FEP) foils in a thermoforming process[12] allowing a variety of sample specific configurations (Fig. 1f, Extended Data Fig. 4 and Methods). Using this mounting strategy, we can ensure growth and environmental conditions similar to experiments performed in standard plates providing consistency between microscopy data and other experiments.

In former work[16], we used the predecessor of the here presented microscope (one detection objective)[17] to track cells in developing intestinal organoids. However, the achieved imaging depth was not sufficient to track cells in larger specimens, including mature organoids. The newly devised approach overcomes this hurdle. We simultaneously imaged crypt and villus formation of maturing mouse intestinal organoids over the course of 3 days (Fig. 1g and Supplementary Video 1), using the cell cycle reporter, FUCCI2 (ref. 18). Dual-color imaging with single-cell resolution within a depth of 360 μm and a temporal resolution of 10 minutes (Supplementary Video 1), allowed the visualization of the in toto dynamics of the organoids in unmatched detail. Visualizing single cells throughout the entire sample volume requires dual-detection. We illustrate this by comparing the xz sections of individual detection objectives to the fused data (Fig. 1h and Methods). The sections of the single views show an expected degradation with increasing imaging depth, whereas the fused data are composed of optimal quality from both views (Supplementary Video 2). This strategy is also necessary to image entire animals such as the cnidarian *Hydra*. We recorded *Hydra* regeneration for 2.5 days starting from newly formed spheroids cut from adult animals[19] and observed the formation of the body axis along with the development of oral and aboral structures, with unmatched temporal resolution (Fig. 1i,j and Supplementary Videos 3 and 4). We also assessed the difference between single and dual-view detection by comparing image quality along increasing imaging depths (Extended Data Fig. 5 and Supplementary Information). The results highlight the importance of dual-detection in large specimens.

To illustrate the versatility of our system, we imaged a variety of samples from 200 to 550 μm in size and for up to 12 days of continuous imaging (Supplementary Table 2 for imaging details of acquisitions): murine liver organoids (Fig. 1k,l and Supplementary Videos 5 and 6), human colon cancer organoids[20] (Extended Data Fig. 6a and Supplementary Videos 7 and 8), murine parotid salivary gland organoids (Extended Data Fig. 6b and Supplementary Video 9) and gastruloids

(Extended Data Fig. 6c and Supplementary Video 10). Additionally, our chamber design allows us to perform parallel perturbation experiments (Supplementary Video 11 and Supplementary Information).

After establishing that the microscope enables both long-term and highly dynamic imaging while allowing multiposition imaging (Supplementary Video 12), we characterized its capabilities to obtain high-quality single-cell data. We performed live imaging of intestinal organoids expressing FUCCI2 (ref. 18) and performed endpoint fixation and immunofluorescence assessing their cell type composition (Fig. 2a)[16]. Due to minimal movement of the sample even after immunofluorescence staining, we overlaid the last live-imaging time point with the stained organoids via 3D registration (Extended Data Fig. 6d and Methods). The Paneth cell marker Lysozyme (Lys) and the secretory cell marker Dll1 were used to detect cells of interest. Triple positive cells ($hCdt1^+/Lys^+/Dll1^+$) were back-tracked to monitor Paneth cell maturation and their cell cycle arrest in G0/G1. As expected, the initial position of the maturing Paneth cells was predictive of the eventual position of the organoid crypt (Fig. 2b,c). Next, we compared $hCdt1^+/Lys^+/Dll^+$ Paneth cells with $hCdt1^+/Lys^-/Dll^-$ cells in the crypt (predominantly intestinal stem cells) and the villus of the organoid (mostly enterocytes) (Fig. 2d). We further identified enterocytes and Paneth cells, which were already terminally differentiated before the recording began. For other cells, especially in the crypt, we identified their moment of emergence, allowing us to track their full maturation with an average cell cycle length of 21.9 h at the time of fixation. Assessing the cell type specific time of emergence, we conclude that Paneth cells emerge earlier than hCdt1 single positive cells suggesting specific cell cycle lengths and an order of specification (Fig. 2e–g and Extended Data Fig. 6e–g). Such insights into cellular behavior and maturation processes would not have been attainable without the combination of live imaging and immunofluorescence within the entire organoid volume.

Unlike most other samples we imaged, gastruloids are dense structures[21] and thus are challenging to image with single-cell resolution. We used this model system to display the capability of our microscope to obtain motility and shape features of individual cells. Standard gastruloid protocols use Wnt activation (Chiron99021, Chir)[21,22], to increase mesoderm formation efficiency. This potentially induces an epithelial to mesenchymal transition-like behavior and an increase in cell migration[23]. To analyze the cell shape within gastruloids, we generated chimeras (Fig. 2h) expressing a membrane reporter (Lck-GFP) in a subset of the cells (~10%). Subsequently, we recorded the dynamics of gastruloids before (42 h after aggregation), during (66 h after aggregation) and after the Wnt pulse (90 h after aggregation) for 5.5 hours with 10 minute intervals (Supplementary Video 13). Additionally, we embedded the gastruloids in 40% Matrigel to prevent mechanical rotations in the sample chambers. 3D segmentation of single cells using Cellpose[24] (Fig. 2i,j) allowed the calculation of major/minor axis ratios, showing an increase in cellular elongation peaking during Chir treatment (Fig. 2k). At later stages (90 h after aggregation),

**Fig. 1 | Open-top dual-view light-sheet microscope with examples of time-lapse acquisitions of various multicellular systems. a**, Model of the dual-view light-sheet microscope, showing the incubator (IN), sample mounting area (SA), transmitted light (TL) and sample positioning unit (PS). Blue and green dashed lines indicate the positions of the cross-sections shown in **b** and **c**. **b**, Cross-section of the model of the microscope showing the illumination objectives (Ill1, Ill2) in blue. **c**, Cross-section of the model of the microscope showing the two detection objectives (Det1, Det2) in green. **d**, Top view showing the two detection (detection 1 and 2) and two illumination objectives (illumination 1 and 2) with space for sample chambers. The green arrows indicate the direction of the emission light; the blue arrows indicate the excitation light. **e**, Side view of the objective arrangement. The zoom-in shows an image with the illumination beams and the bottom of a sample chamber. **f**, Model of the sample holder with four different sample chambers. **g**, Maximum intensity projections (MIPs)

along the z axis showing three time points from an acquisition with organoids expressing the Fucci2-reporter (hGem-mVenus and hCdt1-mCherry). The yellow line indicates the position of the cross-section in **h**. Scale bar, 50 μm. **h**, Cross-section in xz plane of the intestinal organoid in **g** using detection 1, detection 2 and the fused data. Scale bar, 50 μm. **i**, MIPs along the z axis showing three time points from time-lapse acquisition of *Hydra* expressing an ectoderm-reporter (ecto [β-act::RFP]). The yellow line indicates the position of the cross-section in **j**. Scale bar, 50 μm. **j**, Cross-section in xz plane of the *Hydra* shown in **i** using detection 1, detection 2 or the fused data. Scale bar, 50 μm. **k**, MIPs along the z axis showing three time points from a time-lapse acquisition of liver organoids expressing mg-GFP and H2B-mCherry. Scale bar, 50 μm. **l**, 3D rendering of the liver organoid shown in **k**. The zoom-in shows a region with and without membrane signal. Scale bar, 50 μm.

a subpopulation of cells displayed long cell protrusions (Fig. 2i). This observation led to the hypothesis that cell motility is increasing along gastruloid development. Using the Fiji plugin Mastodon[25] we tracked Lck-GFP[+] (green fluorescent protein positive) cells (Fig. 2l,m) and found that the median velocity of migration ($\mu$m h$^{-1}$) increased along gastruloid development. The strongest acceleration occurred during the Wnt pulse exhibiting a 1.4-fold increase in median speed compared to the time window before the Wnt pulse (Fig. 2n). Gastruloids imaged post-Wnt activation exhibited the longest track length (Fig. 2o).

Evaluation of the 3D mean-square displacement (m.s.d.$_{3D}$) suggested an increased speed and a change in migration behavior of cells tracked from 90 h onward (Fig. 2p). Together, these findings demonstrate an increase in migration, suggesting that Wnt activation enhances cell motility and potentially coordinated migration in gastruloids[26]. The observed changes in cell shape, and increased motility, suggest at least a partial epithelial to mesenchymal transition-like transition[21]. This trend remains consistent between gastruloids embedded in Matrigel and gastruloids in suspension (Extended Data Fig. 6h,i).

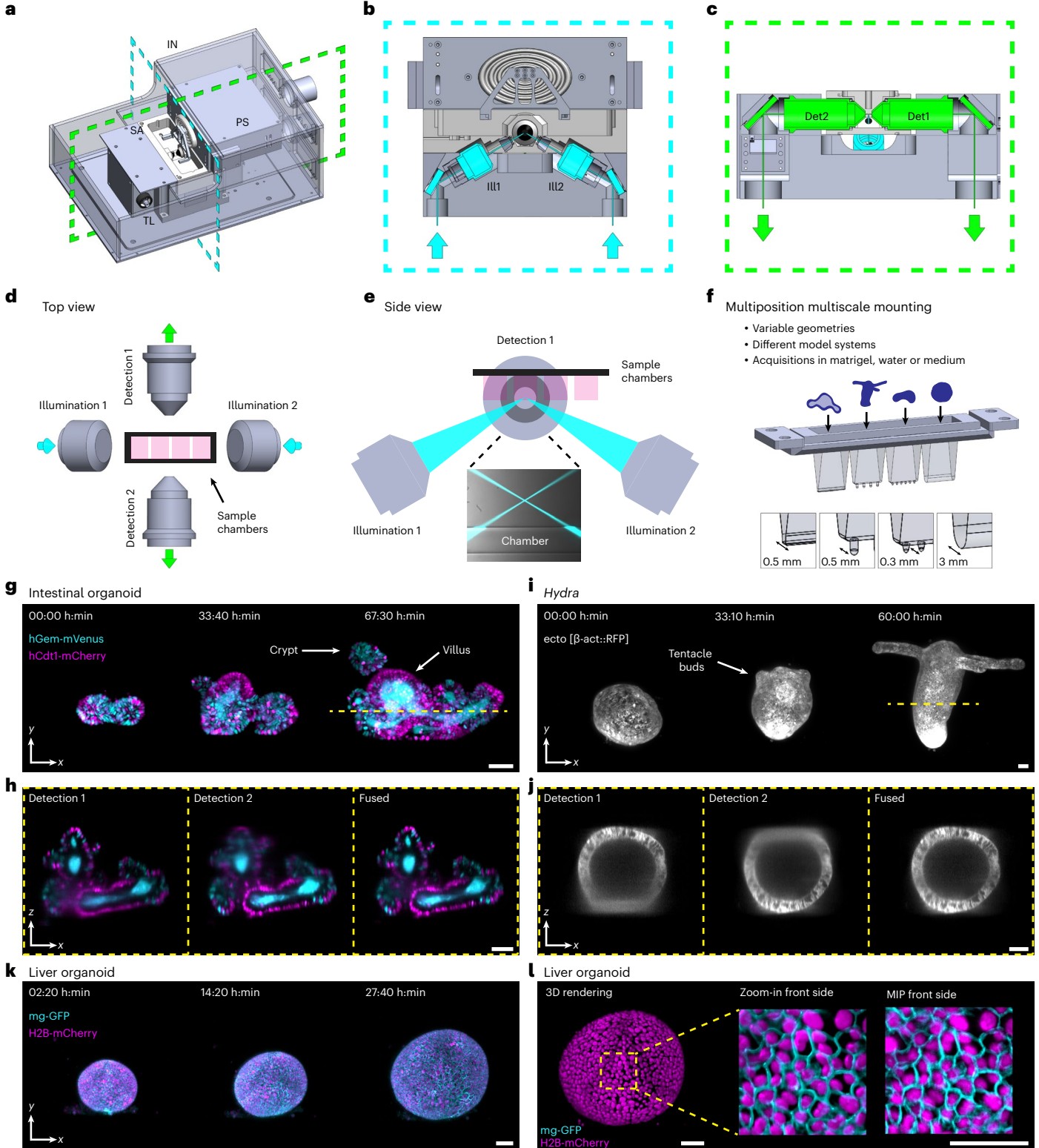

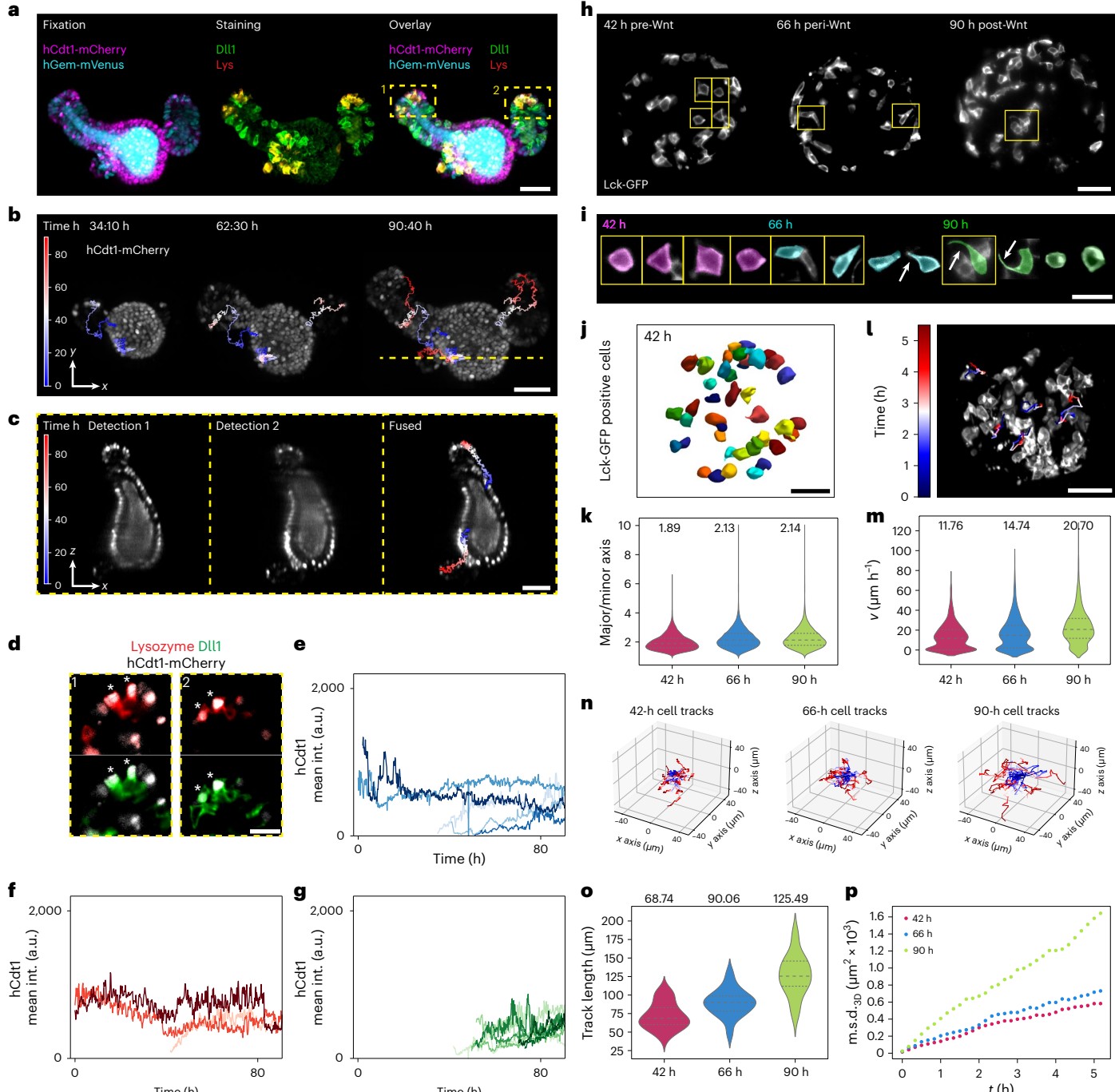

**Fig. 2 | Single-cell analysis of intestinal organoids and gastruloids.**
**a**, Intestinal organoid expressing hGem-mVenus and hCdt1-mCherry (left) stained for Lysozyme (Lys) and DLL1 (middle). If not indicated differently.
**b**, MIPs of stills of an intestinal organoid expressing hCdt1-mCherry. Overlaid are the tracks of back-tracked cells color coded over time. The dashed line corresponds to the z projections shown in **c. c**, The z projections of the intestinal organoid shown in **b** of detection 1, detection 2 and the fused data. Overlaid are tracks from back-tracked cells showing temporal progression. **d**, Zoom-ins of intestinal crypts shown in **a**. Asterisks indicate triple positive cells for hCdt1+/Dll1+/Lysozyme+. **e**–**g**, Quantification of hCdt1, Lysozyme and Dll1 intensities over time for individual cells at indicated final positions: hCdt1, Lys, Dll1 positive crypt cells (**e**); hCdt1-positive villus cells (**f**) and hCdt1-positive crypt cells (**g**). **h**, The z planes of gastruloids expressing Lck-GFP at three time points (42 h pre-Wnt, 66 h peri-Wnt and 90 h post-Wnt). **i**, Individual cells partly highlighted in **h** from gastruloids 42, 66 and 90 h after cell seeding.

Arrows indicate cell protrusions. **j**, Representative 3D segmentation (Cellpose) of Lck-GFP positive cells of gastruloid 42 h after cells seeding. **k**, Comparison of the major/minor axis ratio 42, 66 and 90 h after seeding, showing the median (values depicted in figure) and the first and third quartile. **l**, MIPs of a gastruloid overlaid with the tracked cells over time. **m**, Violin plots of cell velocity (μm h⁻¹) grouped by observation windows, showing the median (values depicted in figure) and the first and third quartile. **n**, Cell tracks of individual cells of gastruloids imaged for 5.5 h mounted at 42, 66 and 90 h postseeding centered at the origin of the coordinate system, color coded for the temporal progression. **o**, Track length per cell for the individual imaging windows of Lck-GFP gastruloids. Violin plots of cell velocity (μm h⁻¹) grouped by observation windows, showing the median (values shown in figure) and the first and third quartile. **p**, The m.s.d. averaged over all cells for the individual imaging windows of Lck-GFP chimera gastruloids. Scale bars, 50 μm, **a**–**d**,**h**,**j**,**l**; 20 μm (**i**).

In summary, we present a dual-view and dual-illumination open-top light-sheet microscope suitable for long-term multiposition imaging of a wide range of samples at single-cell resolution and with quality suitable for cell segmentation and tracking of cells in the entire organoid. Sample specific and flexible mounting was achieved by using a comparatively[27] simple thermoforming process to manufacture sample holders of various shapes.

## Discussion

Previously, different open-top light-sheet microscopes have been developed to combine the 3D imaging capabilities of light-sheet microscopy with multiwell plates[9,11,13–15,28–31]. Another solution proposed the use of two opposing illumination objectives to minimize shadowing effects[17]. None of these approaches offers detection from two opposing sides. On the contrary, multiview light-sheet microscopy was developed[4–6,32]. However, these approaches are limited in throughput due to a constraint geometry or require sample embedding, which is not always suitable. Our system combines the advantages of an open-top geometry with a multiview approach.

In the future, this microscope will be a promising platform for further technical advancements, such as laser ablation or optogenetic stimulations. Integration of adaptive optics or imaging of optically cleared specimens could further enhance image quality.

## Online content

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

## Methods

All animal experiments are approved by the Basel Cantonal Veterinary Authorities and conducted in accordance with the Guide for Care and Use of Laboratory Animals.

Patient samples were collected with freely given, specific, informed and unambiguous consent and ethics approval for organoid derivation and data collection (Biobanks Review Committee, UMC Utrecht, subbank numbers 12/093). Patient-derived organoids identified by the HUB code P-19bT CRC organoids are cataloged at www.huborganoids.nl and were requested at techtransfer@huborganoids.nl. Distribution to third (academic or commercial) parties can be authorized by the Biobank Research Ethics Committee of the University Medical Center Utrecht (TCBio) at request of Hubrecht Organoid Technology (HUB).

### Microscope

The presented light-sheet microscope consists of two illumination paths, two detection paths and a transmitted light beam path. For illumination 60 mW 488 nm (LuxX 488-60), 80 mW 515 nm (LuxX 515-80), 50 mW 561 nm (OBIS 561-50) and a 100 mW 638 nm (LuxX 638-100, all lasers from Omicron-Laserage Laserprodukte) laser are used. Lasers are combined in a laser combiner (LightHUB+, Omicron-Laserage Laserprodukte) and coupled into a single-mode optical fiber with a 0.7 mm collimated beam output.

The collimated illumination light from the fiber is first reflected by a mirror (BB05E02, Thorlabs) mounted in a kinematic mount (POLARIS-K05S2, Thorlabs) and passed through a filter wheel (FW212C, Thorlabs) containing neutral density filters (NE510B, NE520B, NE530B, Thorlabs) to further attenuate the intensity by a factor of 10, 100 or 1000.

The laser beam is reflected by a system of four galvanometric mirrors (6210H, Novanta Cambridge Technology) that are placed in custom-made aluminum mounts. By a compound movement of the four scanners, the beams can be translated and rotated on the image plane ($xy$) to generate light-sheets by scanning as well as being translated and rotated in $yz$ plane for focus adjustment. After passing through a scan lens made from two achromatic lenses (47–718, Edmund Optics), the illumination beam passes through a tube lens made from two achromatic lenses (49–281 and 49–283, Edmund Optics) followed by a custom triplet lens (plano-concave, $r = 41.67$, 7980-0F, biconvex, $r = 49.87$, S-TIH4, plano-concave, $r = 41.67$, S-TIH53, Optimax) and a 10× air objective with an NA of 0.2 (T Plan EPI SLWD 10×, Nikon) and a glass window. The illumination beam reaches the sample at an angle of 30° with the horizontal axis crossing an air glass and glass water interfaces. The custom-designed correction triplet lens compensates for chromatic and spherical aberrations caused by different media in the light path to achieve diffraction limited resolution.

To switch between the two illumination beam paths, a D-shaped pickoff mirror (PFD10-03-P01, Thorlabs) placed at an angle of 45° after the scan lens is used to reflect the beam either to the light path of illumination 1 or 2. Both illumination paths contain a custom-made retroreflector system that contains two prism mirrors (15–599, Edmund Optics) mounted on a linear stage (UMR25, Newport) to adjust beam collimation at the back focal plane of the illumination objective and axial position of the illumination beam focus.

On the detection side, the fluorescence signal is collected by each of the two ×16 water immersion objectives with an NA of 0.8 (CFI75 LWD ×16 W, Nikon) and a working distance of 3 mm, which oppose each other. The signal is reflected by two mirrors to match the 200 mm focal length of the tube lens (T12-LT-1X, Nikon) and passes through motorized filter wheels (LEP filter wheel, Ludl) each equipped with the following emission filters FF01-515/LP-25, FF01-523/610-25, FF01-542/27-25 (all from Semrock) and ZET405/488/561/640mv2 (Chroma Technologies). The fluorescence images are acquired using ORCA-Fusion sCMOS cameras (C14440-20UP, Hamamatsu). Each camera is mounted together

with the filter wheel on a custom aluminum mount that is placed on a manually adjustable liner stage (M-UMR5.25, Newport) to bring the two views in focus. The ORCA-Fusion camera contains a sensor with 2,304 × 2,304 pixels. With this combination of detection objective and camera, we achieve a pixel spacing of 0.406 μm and a field of view of 935 × 935 μm. For this camera and objective configuration, the resolution is limited by pixel sampling (FWHM in Matrigel based on bead measurements is 0.8 μm lateral and 2.9 μm axial; Extended Data Fig. 3).

In one of the detection beam paths, a light-emitting diode light (LED770L, Thorlabs) is reflected by a 750 nm short-pass dichroic mirror (FF750-SDi02, Semrock) to the objective that serves as a condenser to acquire transmitted light images, while the emitted fluorescence light still passes through the dichroic mirror to the camera.

To acquire one sample plane, both cameras must be positioned to have a common focal plane and both light-sheets must be aligned to be in the focal plane of the cameras. The alignment can be done by the following procedure: first, the illumination beams are moved with the galvanometric mirror to be in the focal plane of one camera. Then, the second camera is moved with micrometer screws to be aligned with the illumination beams and focal plane of the first camera.

The light-sheet is generated by scanning of the Gaussian beam within the focal plane. To acquire an image plane, the sample is illuminated within the camera exposure time first from one side and then from the other side. Two cameras acquire the two views simultaneously and the data coming from the two views can be fused after the acquisition to one stack. To generate a 3D stack, the sample is moved and the focal planes and light-sheets are kept in a fixed position.

The microscope was designed using Solidworks 2018 SP 1.0.

### Chamber fabrication

Chambers are produced from FEP using a vacuum thermoforming process. FEP foil (Adtech Polymer Engineering) with a thickness of 127 μm is cut in quadratic sheets of approximately 15 × 15 cm. A sheet of foil is then clamped inside a vacuum forming machine (Jintai JT-18, Yuyao Jintai Machine Factory), where the foil is heated up for 8 min. After the foil has heated up, custom-made aluminum pieces are placed below the foil and serve as molds. Under vacuum, the foil is formed around the molds for 30 s (Extended Data Fig. 4a,b). Chambers are then cut out manually from the thermoformed FEP foil and placed inside the sample holder (Extended Data Fig. 4c).

The chambers fit into the 6 mm space in between the two detection objectives and allow access for pipetting from the top and meeting the requirements of different biological samples. For specimens embedded in a matrix such as Matrigel, we developed chambers with a straight bottom (Fig. 1f) with a small width to minimize degradation of image quality caused by Matrigel. For samples grown in suspension such as *Hydra* or gastruloids, we designed chambers with pocket sizes adapted to the size of the specimens. Since the sample holder and the molds for the imaging chambers can be produced easily with 3D printing or aluminum milling, a wide range of different geometries are possible, giving maximal flexibility.

### Chamber mounting

To mount the chambers into the microscope, individual thermoformed chambers are fixed in a 3D printed sample holder that has room for four chambers. This holder is placed onto a *xyz* motorized sample positioning stage assembled by combining three piezo stages (2× CLS 3232-S, 1× CLS 3282-S, SmarAct). This system allows a maximal travel distance across the chambers of roughly 50 mm (long axis). Since water immersion objectives are used for detection, the chambers are lowered into a water reservoir with the objectives immersed below the water surface. The water reservoir and the sample stage are covered by a lid to minimize water evaporation. The sample handling area of the microscope is inside an incubator box to ensure an environmentally controlled area for temperature (CUBE2, Life Imaging Services) and $CO_2$

(LS2 Live gas controller, Viventis Microscopy). To ensure optimal temperature and $CO_2$ concentration, both values are measured in the sample area. For temperature, a pt100 probe (TF101P-1m with GMH 3710, Greisinger) is placed inside the immersion water about 10 mm from the sample and for $CO_2$ concentration a sensor (GC-0006, CO2Meter) is used to measure $CO_2$ concentration in the sample area.

## Microscope control software and electronics
All parts of the microscope are controlled by a microscope control software (Viventis Microscopy). The controller and sensor module of the positioning system (MCS2-MOD, MCS2-S, SmarAct) and the driver of the galvanometric scanners (673, Novanta Cambridge Technology) are powered and connected according to the manufacturer's instructions in a custom electronics enclosure. Digital and analog signals to control lasers and galvanometric scanners are generated by an field-programmable gate array-based real-time controller (Viventis Microscopy).

## Microscope used for benchmarking experiments
For the comparison of the microscope presented in this work, a microscope system as described in refs. 16,17, with the following configuration, is used: for excitation the light-sheet microscope is equipped with a 488 nm (LuxXPlus 488-60), a 561 nm (OBIS 561-50) and a 630 nm (LuxXPlus 630-150) laser. For illumination, two ×10 water immersion objectives with an NA of 0.3 (CFI Plan Fluor 10XW, Nikon) are installed. The light-sheet is generated by scanning the laser beam with a galvanometric scanner system and has a thicknesses (FWHM) of approximately 2.2 μm. A ×25 1.1 NA objective (CFI75 Apo 25XW; Nikon) is used for detection. The images are acquired by an sCMOS camera (Zyla 4.1, Andor). Before the camera, a filter wheel is placed offering the following filters: 488 LP Edge Basic Longpass Filter, F76-490; 561 LP Edge Basic Longpass Filter, F76-561 and HC Dualband Emitter R 488/568, F72-EY2, Semrock, AHF.

## Image processing
For fusion of the data obtained from the two detection objectives, the stacks coming from the two detection objectives are registered to perfectly overlap by using the open-source Python imaging library DIPY. A rigid-body transformation can be used to compensate for small mechanical misalignments between the two detection objectives.

Second, to fuse the data coming from the two detection objectives, the optimal $z$ plane is identified to switch from one view to the other to ensure the highest possible image quality in the fused image stack. Therefore, for both image stacks an image quality score is calculated plane by plane based on the Shannon entropy of the normalized discrete cosine transform as discussed in ref. 33. The metric allows to set the switching $z$ plane to that point, where the opposing detection objective shows the higher image quality.

The stacks are then fused using a sigmoidal function centered at the switching plane and a constant intensity offset is subtracted to compensate for the background of the cameras. The code for data fusion is available in the GitHub repository: https://github.com/fmi-basel/gliberal-lightsheet-2023.

Image data are visualized using ImageJ v.2.9.0 and Paraview v.5.10.1.

## Light-sheet characterization
To evaluate the properties of the light-sheet, images with the static beams were acquired. The images of the static beams were rotated in Fiji such that the orientation of the static beam aligns with the horizontal axis. Subsequently, line profiles were generated and the obtained intensity profile was fitted with a Gaussian function:

$$f(x) = a + b \times e^{\frac{-(x-\mu)^2}{2 \times \sigma^2}}$$

with $a$ being a constant offset, $b$ a scaling parameter, $\mu$ the mean of the function and $\sigma$ the standard deviation. The FWHM was calculated with FWHM $= 2 \times \sqrt{2 \times \ln(2) \times \sigma}$.

With this approach, the beam width $w_0$ and Rayleigh length $z_r$ were determined.

The effective NA of the illumination objectives was calculated using the beam width with

$$NA_{eff} = \frac{n\lambda}{\pi w_0}$$

where $n$ is the refractive index and $\lambda$ the emission wavelength.

## PSF quantification
The point spread function measurement was performed in 50% Matrigel diluting fluorescent beads (Invitrogen TetraSpeck Microspheres, 0.1 μm T7279) in a concentration of 1:1,000. Stacks in different positions within the sample chamber were acquired using an isotropic pixel spacing ($0.406 \times 0.406 \times 0.406$ μm). For illumination, a 488 nm laser was used and emission was collected in the green spectrum. The FWHM in the $xy$ and $xz$ plane was determined by fitting the intensity distribution in the corresponding planes with the Gaussian function as described above. The average FWHM lateral and axial was determined by averaging the quantifications of several beads in different depths inside the sample chamber.

## Single-cell tracking
Single cells of the intestinal organoid and the gastruloids were tracked manually by making use of the Fiji Plugin Mastodon v.1.0.0-beta 26 (ref. 25). For the intestinal organoids, we used the numerical feature extraction of Mastodon to compute the mean intensity of all the spots marking the position of the cells. The mean intensities together with the spot positions for each track was exported as a .csv data table.

For the tracked cells of the gastruloid, we exported only the spot positions for each track as a .csv data table.

The analysis of the cell tracks was done by custom-written Python scripts relying on functions from open-source Python libraries numpy, pandas, seaborn and matplotlib.

The m.s.d.$_{3D}$ for the trajectories $r_i(t)$ of the gastruloid cells labeled with index $i$ was calculated as follows[34]:

$$m.s.d._{3D}(t) = \frac{1}{N} \sum_{i=1}^{N} (r_i(t) - r_i(0))^2$$

where $N$ is the number of cells and $r_i(0)$ the initial position of the cell.

## Single-cell segmentation
Single-cell segmentation of the gastruloids was performed with Cellpose 2.0 v.2.2 (ref. 24). To train the model, a stack from a gastruloid 42, 66 and 90 h after cell seeding was used. Each plane of the stacks was manually annotated, and all three stacks were used to train a model for cell segmentation. To generate the masks in 3D we used the two-dimensional prediction for image plane, which were then stitched together based on the overlap of the masks.

To extract features from the 3D segmentation we relied on the 3D feature extraction from Python library scikit-image using the version v.0.20.0.dev0. The extracted features were analyzed with custom-written Python scripts based on the open-source libraries numpy, pandas, seaborn and matplotlib.

## Sample preparation
All culturing methods and sample preparation steps are described in detail in the Supplementary Methods and Supplementary Tables 2 and 3.

## Statistics and reproducibility

For all experiments, no statistical method was used to predetermine sample size. No data were excluded from the analyses, except for the analysis of cell shapes based on segmentations. False segmentations were excluded by the cell volume and values for major and minor axis. The experiments were not randomized. The investigators were not blinded to allocation during experiments and outcome assessment.

In Fig. 1g, the experiment was repeated twice acquiring 10 and 12 organoids in parallel. In Fig. 1i the experiment was repeated twice acquiring five and six *Hydra* in each acquisition. In Fig. 1k the experiment was repeated twice imaging eight and nine organoids in parallel.

In Fig. 2e $n = 5$, Fig. 2f $n = 3$ and Fig. 2g $n = 8$ cells were investigated. In Fig. 2k, measurements were performed over 33 time points on the 3D volumes of three gastruloids per time window in three different experiments (in total 27 gastruloids). Summing up individual time points, a total of 14,356 datapoints (42 h), 19,149 datapoints (66 h) and 21,495 datapoints (90 h) were analyzed. In Fig. 2m, the following numbers of datapoints (from nine individual gastruloids) were tracked per observation window: $n = 1,983$ (42 h), $n = 1,792$ (66 h) and $n = 1,728$ (90 h). In Fig. 2n, tracks were generated for 24 cells over 32 time points (42 h), 19 cells over 33 time points (66 h) and 18 cells over 32 time points (90 h). In Fig. 2o, the track length was measured for 62 cells (42 h), 56 cells (66 h) and 42 cells (90 h). In Fig. 2p, m.s.d. measurements were performed over 32 time points for 62 cells (42 h), 56 cells (66 h) and 42 cells (90 h).

### Reporting summary

Further information on research design is available in the Nature Portfolio Reporting Summary linked to this article.

## Data availability

A representative subset of the imaging data for each model system presented in the figures and videos is available on figshare. The full datasets are available on request. Source data are provided with this paper.

## Code availability

Codes used to generate the findings of this study are available on the publicly available GitHub repository https://github.com/fmi-basel/gliberal-lightsheet-2023.

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

## Acknowledgements

We thank M. Klement (Department of Biosystems Science and Engineering) for producing sample holders and chamber molds and E. Tagliavini for IT support. We acknowledge V. Kalck for general support, organoid cultivation and genotyping of mouse strains. C. Azzi and M. Hartl contributed to the mouse embryonic stem cell line engineering by providing plasmids and performing parts of the cloning work. R. Verhagen helped with cell tracking. This project received funding from the SNSF (The Swiss National Science Foundation) Sinergia grant (no. CRSII5_189956, P.L.) and the European Research Council under the European Union's Horizon 2020 research and innovation program (grant agreement no. 758617, P.L.).

## Author contributions

P.L. and P.S. supervised the work. P.L., P.S. and A.B. originally conceived the project. F.M., G.d.M., A.B., C.R. and P.S. designed and constructed the microscope. P.S. wrote the microscope software. F.M. performed experiments and developed the data analysis software supported by G.d.M. S.S. performed experiments and contributed to analyzing the data. K.C.O. contributed to organoid culturing and performing experiments. S.L.W. and C.T. contributed to the *Hydra* experiments. F.M., S.S., G.d.M., P.S. and P.L. wrote the manuscript.

## Competing interests

A.B. and P.S. are cofounders of Viventis Microscopy Sàrl that commercializes the light-sheet microscope presented in this work (patent pending). The other authors declare no competing interests.

## Additional information

**Extended data** is available for this paper at https://doi.org/10.1038/s41592-024-02213-w.

**Correspondence and requests for materials** should be addressed to Petr Strnad or Prisca Liberali.

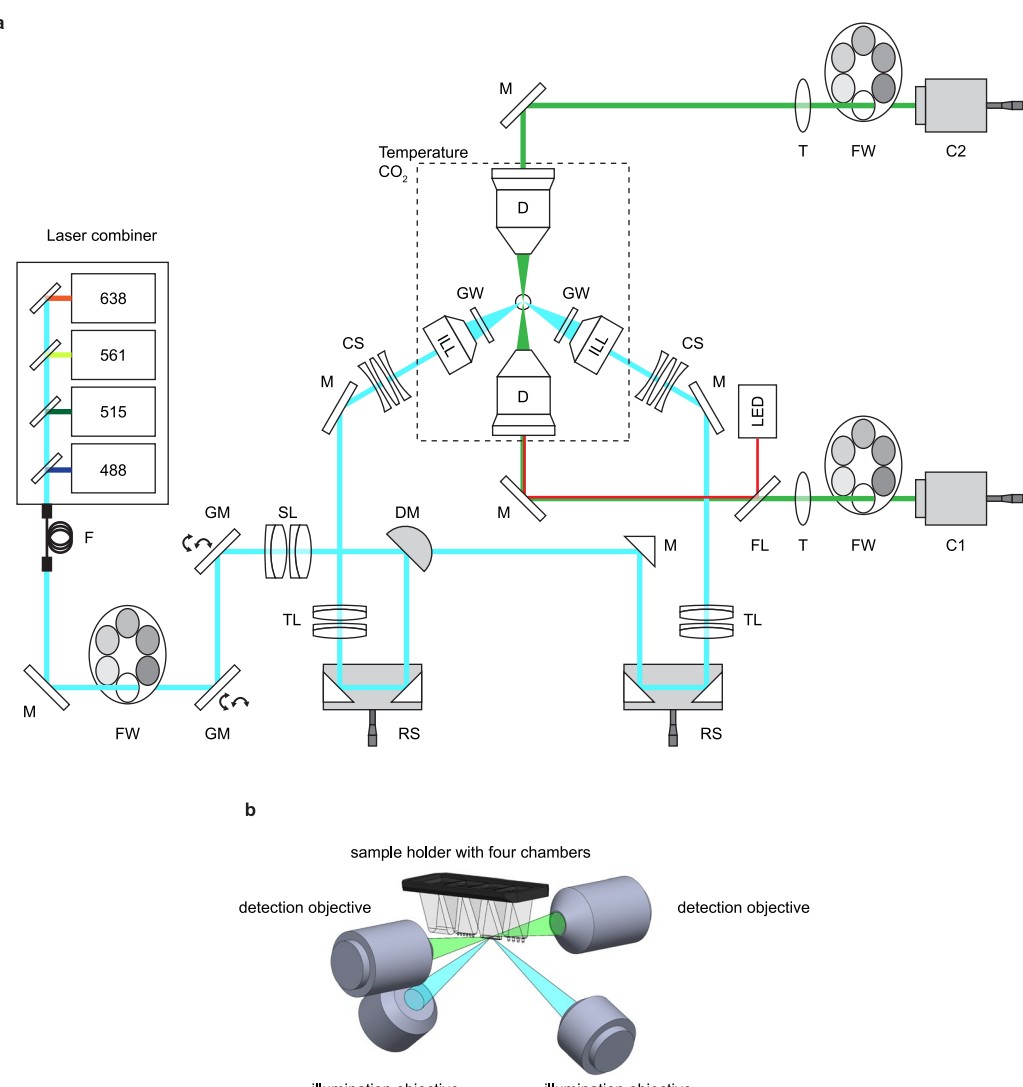

**Extended Data Fig. 1 | Light path and objective area of the dual-view light-sheet microscope. a**) Schematic representation of the light path of the dual-view and dual-illumination light-sheet microscope: fiber (F), mirror (M), filter wheel (FW), galvanometric mirror (GM), scan lens (SL), d-shaped mirror (DM), reflector system (RS), tube lens (TL, T), correction system (CS), illumination objective (ILL), glass window (GW), detection objective (D), dichroic mirror (FL), camera 1 and 2 (C1,C2). See Methods section. **b**) 3D schematic of the objective arrangement and the position of the sample holder.

**a** Open top multi sample dual view light sheet

**b** Single objective light sheet

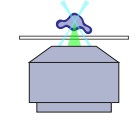

Dunsby 2008
Bouchard et al. 2015
Yang et al. 2019
Millett-Sikking et al. 2019

**c** Single objective multi view light sheet

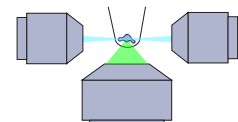

Sparks et al. 2020
Yang et al. 2022

**d** Open top light sheet

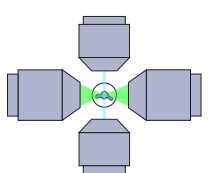

McGorthy et al. 2015
Strnad et al. 2015
Glaser et al. 2019

**e** Single objective light sheet in micro cavities

Galland et al. 2015
Beghin et al. 2022

**f** Open top dual illumination single detection light sheet

Serra et al. 2019

**g** Multiview light sheet

Tomer et al. 2012
Krzic et al. 2012
Schmid et al. 2013
McDole et al. 2018

**h**

| Concept | Multi positioning | Multi well | Open top geometry | Dual illumination | Dual view | Opposing detection views | Sample rotation |
|---|---|---|---|---|---|---|---|
| Open top multi sample dual view light sheet | Yes | Yes | Yes | Yes | Yes | Yes | No |
| Single objective light sheet | Yes | Yes | Yes | No | No | No | No |
| Single objective multi view light sheet | Yes | Yes | Yes | Yes | Yes | No | No |
| Open top light sheet | Yes | Yes | Yes | No | No | No | No |
| Single objective light sheet in micro cavities | Yes | Yes | Partially | Yes | No | No | No |
| Open top dual illumination single detection light sheet | Yes | Yes | Yes | Yes | No | No | No |
| Multiview light sheet | Limited | No | No | Yes | Yes | Yes | Yes |

**i**

| Concept | Reference | Lateral Resolution [um] | Axial Resolution [um] | Fiel of View [um x um] | Detection NA |
|---|---|---|---|---|---|
| Open top multi sample dual view light sheet | This work | 0.8 | 2.9 | 935 x 935 | 0.8 |
| Single objective light sheet | Dunsby 2008 | 0.82 | N.A. | 81 x 81 | 0.45 |
| | Yang et al. 2019 | 0.34 | 0.6 | 100 x 70 | 1.27 |
| Single objective multi view light sheet | Sparks et al. 2020 | 0.5 | 1.2 | 295 x 295 | 0.93 |
| | Yang et al. 2022 | 0.48 | 1.8 | 750 x 750 | 1.0 |
| Open top light sheet | McGorthy et al. 2015 | 1.1 | 6.2 | 583 x 583 | 0.3 |
| | Strnad et al. 2015 | N.A. | N.A. | 130 x130 | 1.1 |
| | Glaser et al. 2019 | 0.96 | 3.55 | 900 x110 | 0.4 |
| Single objective light sheet in micro cavities | Galland et al. 2015 | N.A. | N.A. | 25 x 25 264 x 264 | 0.3 - 1.4 |
| Open top dual illumination single detection light sheet | Serra et al. 2019 | N.A. | N.A. | N.A. | 1.1 |
| Multiview light sheet | Tomer et al. 2012 | 0.39 | 1.59 | N.A. | 0.8 |
| | Krzic et al. 2012 | 0.272 / 0.374 | 0.54 / 1.02 | 520 x 438 | 1.1 / 0.8 |
| | Schmid et al. 2013 | N.A. | N.A. | 1200 x 1200 | 0.3 |
| | McDole et al. 2018 | N.A. | N.A. | 0.4 | 0.8 / 1.1 |

**Extended Data Fig. 2 | See next page for caption.**

**Extended Data Fig. 2 | Schemes and comparison of key parameters among state-of-art light-sheet microscopy methods. a**) Scheme of the open top multi-sample dual-view light-sheet microscopy (presented in this work). **b**) Scheme of single objective light-sheet microscopy. **c**) Scheme of single objective multi-view light-sheet microscopy. **d**) Scheme of open top light-sheet microscopy. **e**) Scheme of single objective light-sheet microscopy in micro cavities. **f**) Scheme of open top dual-illumination single detection light-sheet microscopy. **g**) Scheme of multi-view light-sheet microscopy. **h**) Comparison of key aspects of different state-of-art light-sheet microscopy methods. **i**) Comparison of technical specifications of state-of-art light-sheet microscopy methods.

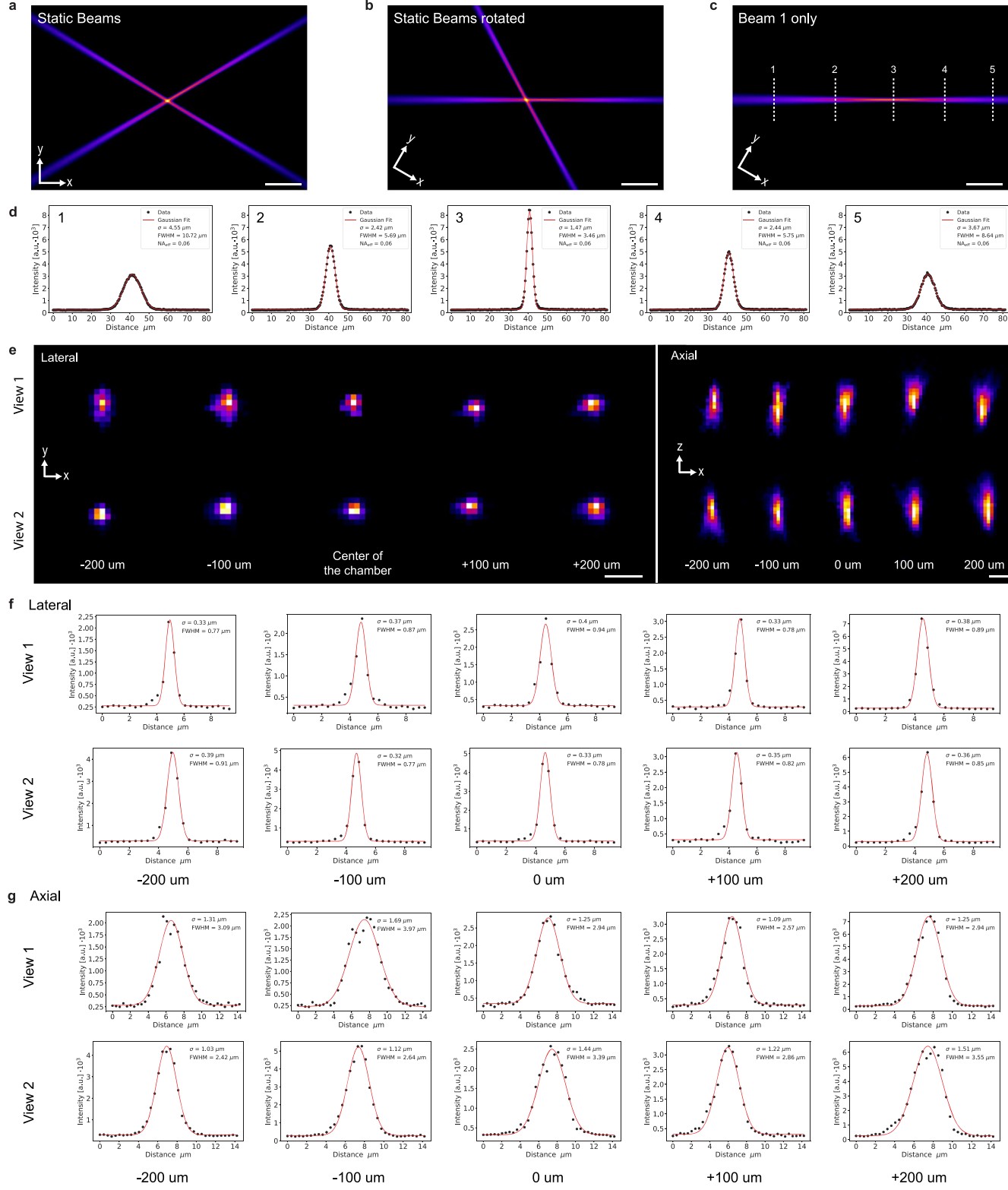

**Extended Data Fig. 3 | Characterization of the light-sheet microscope.**
**a**) Image of the static beams. Scale bar 50 μm. **b**) Image of the static beams rotated in Fiji such that one beam matches the horizontal axis. Scale bar 50 μm. **c**) One single static beam rotated in Fiji. The dashed lines mark the positions of the line profiles shown in d). Scale bar 50 μm. **d**) Quantification of the width of the static beam. The intensity distribution was fitted with a Gaussian function and the FWHM and the effective illumination NA was extracted (Methods). **e**) Imaged PSFs in XY and XZ sections in 5 different depths inside a FEP chamber filled with 50% Matrigel. The 0 μm position marks the center of the chamber (Methods). **f**) Quantification of the XY sections of the PSF shown in e) and extraction of their FWHM. **g**) Quantification of the XZ sections of the PSF shown in e) and extraction of their FWHM.

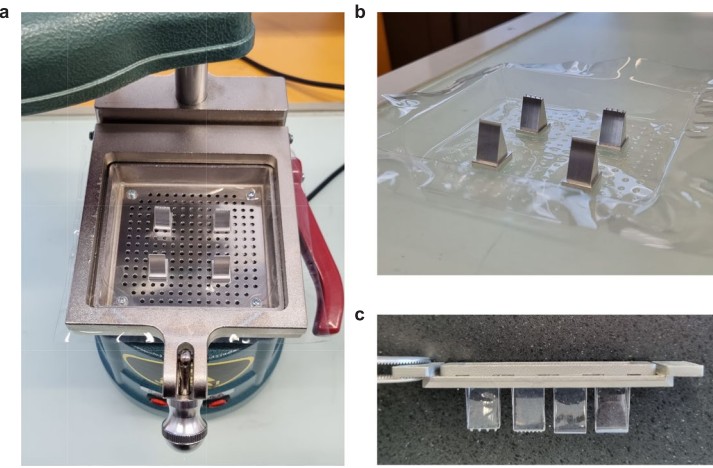

**Extended Data Fig. 4 | Production of the FEP chambers. a**) Four aluminum molds placed in the thermoforming machine. The photograph shows the foil after being successfully thermoformed around the molds. **b**) FEP foil with molds taken out of the machine after processing. **c**) Sample holder with four different FEP chambers fixed inside.

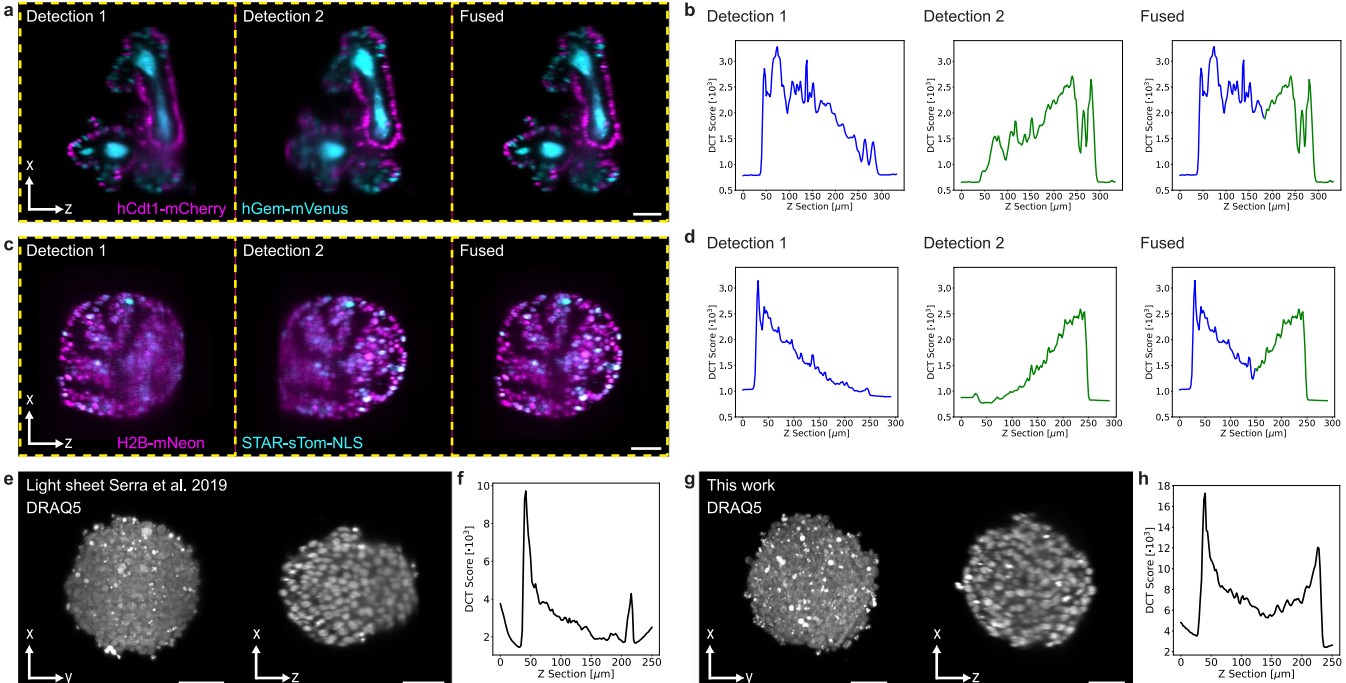

**Extended Data Fig. 5 | Comparison of the image quality using different light-sheet microscopes. a)** Cross section in XZ plane of the intestinal organoid shown in Fig. 1g using Detection 1, Detection 2 and the fused data from both objectives. Scale bar 50 μm. **b)** Comparison of the image quality using the Shannon Entropy of the Discrete Cosine Transform (DCT) as metric for Detection 1, Detection 2 and the fused data. The DCT is calculated for each z section of the image stack corresponding to a. **c)** Cross section in XZ plane of a human colon cancer organoid using Detection 1, Detection 2 and the fused data from both objectives. The experiment was repeated two times with 7 and 8 organoids imaged. Scale bar 50 μm. **d)** Comparison of the image quality using the Shannon Entropy of the DCT as metric for Detection 1, Detection 2 and the fused data. The DCT is calculated for each z section of the image stack corresponding to c. **e)** Maximum intensity projection (MIP) along the Z axis and a cross section in the XZ plane of a gastruloid stained with the nuclear marker DRAQ5 imaged with the light-sheet microscope published in[17]. In total n = 9 gastruloids were imaged. Scale bar 50 μm. **f)** Image quality plot using Shannon Entropy of the DCT along individual z sections of the image stack corresponding to e. **g)** Maximum intensity projection (MIP) along the Z axis and a cross section in the XZ plane of a gastruloid stained with the nuclear marker DRAQ5 imaged with the light-sheet microscope presented in this work. In total n = 7 gastruloids were imaged. Scale bar 50 μm. **h)** Image quality plot using Shannon Entropy of the DCT along individual z sections of the image stack corresponding to g.

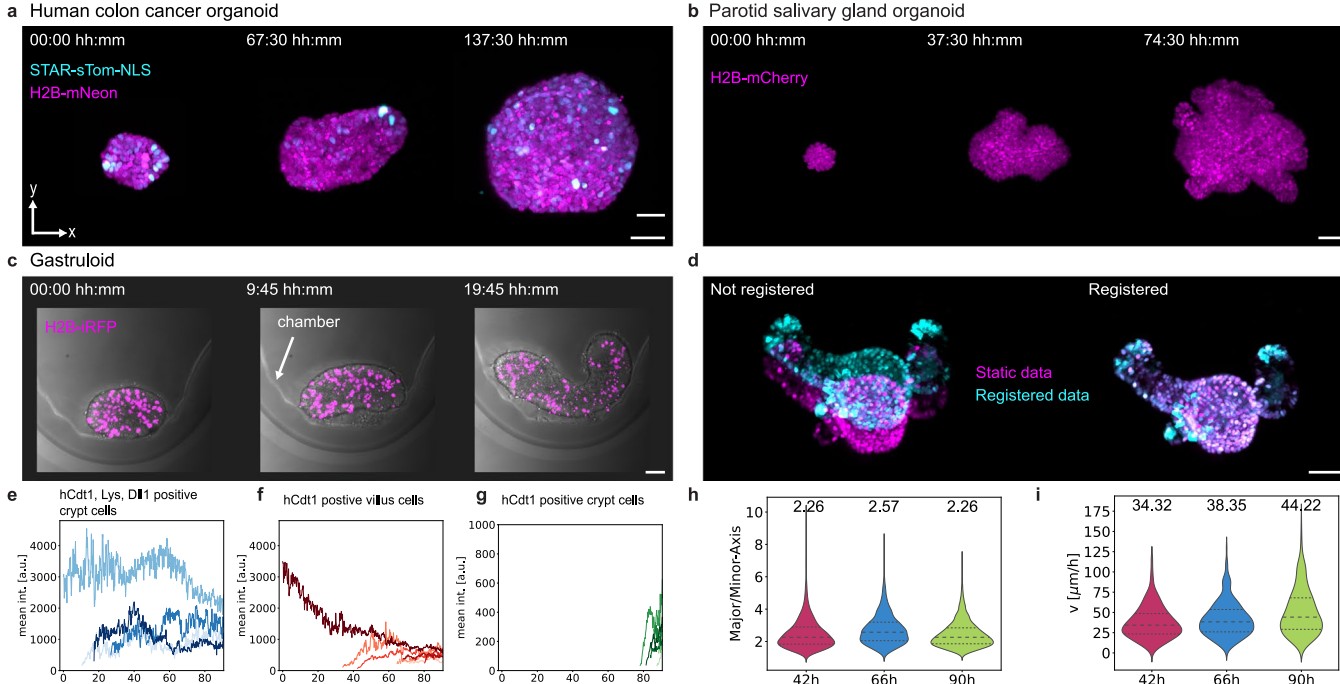

**Extended Data Fig. 6 | Showcase acquisitions, an example of image registration and results of the single cell analysis of gastruloids in suspension. a**) MIPs along Z axis showing three time points from time-lapse acquisition of human colon cancer organoids expressing STAR-sTom-NLS and H2B-mNeon. The experiment was repeated two times with 7 and 8 organoids imaged. Scale bar 50 µm. **b**) MIPs of a parotid salivary gland organoid expressing H2B-mCherry at three different time points of a time lapse acquisition for around 3 days. The experiment was repeated two times with 6 and 7 organoids imaged. Scale bar 50 µm. **c**) MIPs of a gastruloid expressing H2B-iRFP at three different time points of a time lapse acquisition over 20 hours shown together with transmitted light. The experiment was repeated two times with 12 and 13 samples imaged. Scale bar 50 µm. **d**) Example of the registration of an organoid, that was additionally fixed and stained after live imaging. Left image shows the overlay of the MIPs before registration and the right image shows the results after registration. Magenta shows the MIP of the last acquired timepoint and cyan shows the MIP after fixation and staining. Scale bar 50 µm. **e**) Quantification of hCdt1 intensity over time for cells that are triple positive (hCdt1, Dll1 and

Lysozyme) and are located in the crypt at the last time point. Each color represents a single cell ((n = 4). **f**) Quantification of hCdt1 intensity over time for hCdt1-positive cells, that are located in the villus at the last time point. Each color represents a single cell (n = 7). **g**) Quantification of hCdt1 intensity over time for hCdt1-positive cells, that are located in the crypt at the last time point. Each color represents a single cell (n = 9). **h**) Comparison of the ratio of major and minor axis for the three different time windows investigated in gastruloids imaged in suspension. Measurements were performed over 33 timepoints on the 3D volumes of 3 gastruloids per imaging window (42 h, 66 h or 90 h). Summing up individual timepoints, a total of 3753 datapoints (42 h), 3340 datapoints (66 h) and 8861 datapoints (90 h) were analyzed. Median (values in figure), first and third quartile are shown. **i**) Violin plot of the velocity of cells imaged at 3 different windows of gastruloid development showing the median (values in figure) and the first and third quartile. The following numbers of datapoints (from 3 individual gastruloids) were tracked per observation window: n = 622 (42 h), n = 562 (66 h), n = 539 (90 h). Gastruloids were imaged in suspension.

# Reporting Summary

## Statistics

For all statistical analyses, confirm that the following items are present in the figure legend, table legend, main text, or Methods section.

| n/a | Confirmed | |
|---|---|---|
| ☐ | ☒ | The exact sample size (*n*) for each experimental group/condition, given as a discrete number and unit of measurement |
| ☐ | ☒ | A statement on whether measurements were taken from distinct samples or whether the same sample was measured repeatedly |
| ☒ | ☐ | The statistical test(s) used AND whether they are one- or two-sided<br>*Only common tests should be described solely by name; describe more complex techniques in the Methods section.* |
| ☒ | ☐ | A description of all covariates tested |
| ☒ | ☐ | A description of any assumptions or corrections, such as tests of normality and adjustment for multiple comparisons |
| ☐ | ☒ | A full description of the statistical parameters including central tendency (e.g. means) or other basic estimates (e.g. regression coefficient) AND variation (e.g. standard deviation) or associated estimates of uncertainty (e.g. confidence intervals) |
| ☒ | ☐ | For null hypothesis testing, the test statistic (e.g. *F*, *t*, *r*) with confidence intervals, effect sizes, degrees of freedom and *P* value noted<br>*Give P values as exact values whenever suitable.* |
| ☒ | ☐ | For Bayesian analysis, information on the choice of priors and Markov chain Monte Carlo settings |
| ☒ | ☐ | For hierarchical and complex designs, identification of the appropriate level for tests and full reporting of outcomes |
| ☒ | ☐ | Estimates of effect sizes (e.g. Cohen's *d*, Pearson's *r*), indicating how they were calculated |

*Our web collection on statistics for biologists contains articles on many of the points above.*

## Software and code

Policy information about availability of computer code

| | |
|---|---|
| Data collection | Microscope control software (Viventis Microscopy), v2.0.0.2 |
| Data analysis | The microscope was designed using Solidworks 2018 SP 1.0.<br>Data analysis and processing was performed with Python 3.10.9 using the following open-source libraries:<br>scipy 1.10.0, seaborn 0.12.2, pandas 1.5.3, tifffile 2021.7.2, scikit-image 0.20.0.dev0, numpy 1.23.5, matplotlib 3.7.0, dipy 1.7.0<br>The code for fusing the data from the microscope and data analyses is available on GitHub: https://github.com/fmi-basel/gliberal-lightsheet-2023<br>For data visualization ImageJ 2.9.0 and Paraview 5.10.1 was used.<br>For cell tracking Mastodon v1.0.0-beta 26 was used (https://github.com/mastodon-sc/mastodon, v1.0.0-beta-26).<br>For cell segmentation Cellpose 2.0 v2.2 was used (https://github.com/MouseLand/cellpose). |

For manuscripts utilizing custom algorithms or software that are central to the research but not yet described in published literature, software must be made available to editors and reviewers. We strongly encourage code deposition in a community repository (e.g. GitHub). See the Nature Portfolio guidelines for submitting code & software for further information.

## Data

Policy information about availability of data

All manuscripts must include a data availability statement. This statement should provide the following information, where applicable:

- Accession codes, unique identifiers, or web links for publicly available datasets
- A description of any restrictions on data availability
- For clinical datasets or third party data, please ensure that the statement adheres to our policy

Source data supporting the findings of this study will be made available by the corresponding authors on request.

## Human research participants

Policy information about studies involving human research participants and Sex and Gender in Research.

| Reporting on sex and gender | N/A |
|---|---|
| Population characteristics | N/A |
| Recruitment | N/A |
| Ethics oversight | N/A |

Note that full information on the approval of the study protocol must also be provided in the manuscript.

# Field-specific reporting

Please select the one below that is the best fit for your research. If you are not sure, read the appropriate sections before making your selection.

☒ Life sciences    ☐ Behavioural & social sciences    ☐ Ecological, evolutionary & environmental sciences

For a reference copy of the document with all sections, see nature.com/documents/nr-reporting-summary-flat.pdf

# Life sciences study design

All studies must disclose on these points even when the disclosure is negative.

| Sample size | No sample size calculation was performed. In general sample sizes were kept as big as practically possible with the described microscopy setup. |
|---|---|
| Data exclusions | For the analysis of cell shapes based on segmentations, false segmentations were excluded based on cell volume and values for major and minor axis. For showcasing the functionality of the presented microscope, representative specimens are shown in figures and videos. |
| Replication | Experiments were repeated multiple times and only results consitent between repetitions are presented in the manuscript. Where applicable, multiple structures and/or cells were used for analysis and quantification. |
| Randomization | No randomization was applied to select the presented data of intestinal organoids, Hydra, hepatic organoids, human colon cancer organoids, or parotid salivary gland orgnoids. Representative organoids were chosen. For cell tracking in intestinal organoids, an organoid with high morphological and cell type compositional complexity was selected. Comparative EMT related quantifications were performed on gatruloids allocated to experimental groups based on experimental time points. In general, sample randomization was not necessary to showcase the features of the light sheet microscope described in this manuscript. |
| Blinding | The same investigators performed data collection and analysis. Therefore no blinding was performed. |

# Reporting for specific materials, systems and methods

We require information from authors about some types of materials, experimental systems and methods used in many studies. Here, indicate whether each material, system or method listed is relevant to your study. If you are not sure if a list item applies to your research, read the appropriate section before selecting a response.

## Materials & experimental systems

| n/a | Involved in the study |
|-----|----------------------|
| ☐ | ☒ Antibodies |
| ☐ | ☒ Eukaryotic cell lines |
| ☒ | ☐ Palaeontology and archaeology |
| ☐ | ☒ Animals and other organisms |
| ☒ | ☐ Clinical data |
| ☒ | ☐ Dual use research of concern |

## Methods

| n/a | Involved in the study |
|-----|----------------------|
| ☒ | ☐ ChIP-seq |
| ☒ | ☐ Flow cytometry |
| ☒ | ☐ MRI-based neuroimaging |

# Antibodies

| | |
|---|---|
| Antibodies used | The following primary antibodies were used in this study: Sheep anti Dll1 (Catalog no. AF3970, RnD Systems) and rabbit anti Lysozyme (Catalog no. A0099, Dako). Donkey anti rabbit Fab fragments conjugated to Alexa 647 and donkey anti goat Fab fragments conjugated to Alexa 488 fluorophores were used as secondary agents (Catalog no. 705-607-003 and 711-547-003, Jackson Immuno Research). |
| Validation | Primary antibodies were validated in a previous study: Serra, D., Mayr, U., Boni, A. et al. Self-organization and symmetry breaking in intestinal organoid development. Nature 569, 66–72 (2019). https://doi.org/10.1038/s41586-019-1146-y. Fab fragments were validated using non primary stained samples (negaive control) and single stained samples (only one staining per sample). Further, the resulting stainigs were compared to controls using conventional secondary antibodies. |

# Eukaryotic cell lines

Policy information about cell lines and Sex and Gender in Research

| | |
|---|---|
| Cell line source(s) | Human: Female patient-derived organoids identified by the HUB code P-19bT CRC organoids are cataloged at www.huborganoids.nl. Organoids were generated using a transposase-based integration method (movieSTAR: Tol2 insulator8xSTAR-min.pLGR5-sTomato-NLS-pA-PGK-H2BmNeonGreen-2A-Puro).<br><br>Mouse: both male and female mice were used to generate organoids. For "FUCCI experiments" organoids were generated from B6/N x R26 Fucci2 (Tg/+) intestines. Organoids were subsequently infected with pGK Dest H2B-miRFP670 (Catalog no. 90237, Addgene. For the remaining organoid based experiments heterzygotic R26-mG/H2B-mCherry mice were used. These mice originated from crosses of R26-mG (C57BL/6J, Muzumdar, M.D., Tasic, B., Miyamichi, K., Li, L. and Luo, L. (2007), A global double-fluorescent Cre reporter mouse. Genesis, 45: 593-605. https://doi.org/10.1002/dvg.20335) and R26-H2B-mCherry (Abe, T., Kiyonari, H., Shioi, G., Inoue, K.-I., Nakao, K., Aizawa, S., and Fujimori, T. (2011). Establishment of conditional reporter mouse lines at ROSA26 locus for live cell imaging. Genesis 49, 579–590).<br><br>mESC lines for gastruloid culture: E14 (male) and CGR8 (male) cell lines are of 129 background and were provided by the laboratory of Matthias Lutolf (Institute of Human Biology, Basel). |
| Authentication | Cell lines used in this study were not authenticated. |
| Mycoplasma contamination | Cell and organoid lines were routinely tested for mycoplasma contamination. No mycoplasma contaminated material was used in this study. |
| Commonly misidentified lines (See ICLAC register) | No commonly misidentified cell lines were used for this study. |

# Animals and other research organisms

Policy information about studies involving animals; ARRIVE guidelines recommended for reporting animal research, and Sex and Gender in Research

| | |
|---|---|
| Laboratory animals | Mouse:<br>For mG/H2B-mCherry organoids heterzygotic R26-mG/H2B-mCherry mice were used. These mice originated from crosses of R26-mG (C57BL/6J, Muzumdar, M.D., Tasic, B., Miyamichi, K., Li, L. and Luo, L. (2007), A global double-fluorescent Cre reporter mouse. Genesis, 45: 593-605. https://doi.org/10.1002/dvg.20335) and R26-H2B-mCherry (Abe, T., Kiyonari, H., Shioi, G., Inoue, K.-I., Nakao, K., Aizawa, S., and Fujimori, T. (2011). Establishment of conditional reporter mouse lines at ROSA26 locus for live cell imaging. Genesis 49, 579–590). Regarding husbandry, all mice have a 12/12 hours day/night cycle. Medium temperature is 22°C and relative humidity is at 50%. Male and female mice with an age between 5 and 7 weeks were used.<br>In all other cases already established organoid lines were used.<br><br>Hydra:<br>This study used regenerating Hydra vulgaris (ecto[β-act::RFP]/endo[β-act::GFP] "Reverse Watermelon"). Tissue pieces of adult Hydra were used to monitor Hydra regeneration and the formation of new intact individuals. Budding stage Hydra (more than two weeks since detachment) were used. |
| Wild animals | This study did not involve wild animals. |

| Reporting on sex | Sex based information was not collected in this study. Hydra used in this study were propagated asexually. |
|---|---|
| Field-collected samples | This study did not involve samples collected from the field. |
| Ethics oversight | Approved by Basel Cantonal Veterinary Authorities and conducted in accordance with the Guide for Care and Use of Laboratory Animals. |

Note that full information on the approval of the study protocol must also be provided in the manuscript.

