## [Peer Review File · Nature Methods]

Peer Review Information

Manuscript Title: Open top multi-sample dual-view light-sheet microscope for live imaging of large multicellular systems

Corresponding author name(s): Petr Strnad, Prisca Liberali

Editorial Notes: None

Reviewer Comments & Decisions:

Decision Letter, initial version:

Dear Prisca,

I hope you've been well since we last met at the Keystone in Feb.

Thank you for submitting your manuscript entitled "Open top multi sample dual view light sheet microscope for live imaging of large multicellular systems". We have given the paper our careful consideration and find it of potential interest. However, we are concerned that sending the current manuscript out to review could lead to unnecessary delays and possibly an undesirable outcome of the review process.

In particular, we found that the manuscript is missing benchmarking against other LSM methods. This would be important to highlight the advance represented by your method compared to other open top microscopes. While a side by side comparison would really strengthen the paper, we think at least a discussion on this topic is necessary.

We therefore invite you to revise your manuscript to address these concerns before we make a final determination on whether to send your manuscript for external peer-review. Please ensure that the revised version is as concise as possible, and that it conforms to our format requirements (see <http://www.nature.com/nmeth> for our Guide to Authors).

We shall hope to receive your revised version as soon as you are able to complete the suggested revisions. If something similar is published in the interim we will have to consider the impact it has on the novelty of the revised manuscript.

If you anticipate a delay of more than four weeks, please let us know. In this event, we will still be happy to reconsider your paper at a later date so long as nothing similar has been accepted for publication at Nature Methods or published elsewhere. In the event of publication, however, the received date would be that of the revised rather than the original version.

If you are not interested in submitting a revised manuscript in the future please let me know immediately so we can close your file. If you have any questions, please contact me. (Please note I am OOO until Sep 05 but will be happy to chat about this further).

Please use the link below when you are prepared to resubmit.

[Redacted]

The above URL links to your confidential home page and associated information about manuscripts you may have submitted, or that you are reviewing for us. If you wish to forward this email to co-authors, please delete the link to your homepage.

Thank you for your interest in Nature Methods.

Sincerely,
Madhura

Madhura Mukhopadhyay, PhD
Senior Editor
Nature Methods

Author Rebuttal to Initial comments

Friedrich Miescher Institute
for Biomedical Research

Madhura Mukhopadhyay
Senior Editor
Nature Methods

Basel, 3rd of September 2023

Submission of Manuscript

Dear Dr. Mukhopadhyay,

We hereby re-submit our manuscript, entitled "Open top dual view light sheet microscope for live imaging of large multicellular systems" by Franziska Moos, Simon Suppinger, Petr Strnad and myself as a Brief Communication to *Nature Methods*.

As discussed in our previous correspondence via E-mail, we appreciate the suggestions to add further efforts on benchmarking our microscope in the manuscript. Because light sheet microscopy employs relatively diverse concepts and instruments, direct quantitative benchmarking experiments would require access to these instruments and are thus not feasible in the limited time frame. Nevertheless, we added a full supplementary figure and table providing a comprehensive theoretical comparison of major concepts used in light sheet microscopy.

Furthermore, we now provide additional measurements of Shannon Entropy of the Discrete Cosine Transform (DCT) for mature small intestinal and colon cancer organoids, comparing the image quality obtained from single vs dual detection light sheet microscopy. In addition, we also performed a benchmarking experiment using 48 h gastruloids comparing the quality of imaging data generated from an open top dual illumination, single detection light sheet microscope (Serra et al., Nature 2019) with the microscope described in our current study. As our microscope combines high image quality with increased sample throughput, we also included an additional supplementary video depicting 25 small intestinal organoids acquired in parallel in a single experiment to illustrate feasibility of multi positioning experiments.

With the additionally added data we hope that you find our manuscript suitable for peer review and publication in *Nature Methods*.

Yours Sincerely,

Prisca Liberali
Group leader at the FMI and SNSF Professor at the University of Basel

Prof. Prisca Liberali
Maulbeerstrasse 66
CH-4058 Basel

T +41 61 697 66 51
F +41 61 697 39 76

prisca.liberali@fmi.ch
www.fmi.ch

Decision Letter, first revision:

Dear Prisca,

I hope you've been well Your Brief Communication, "Open top multi sample dual view light sheet microscope for live imaging of large multicellular systems", has now been seen by 3 reviewers. As you will see from their comments below, although the reviewers find your work of considerable potential interest, they have raised a number of concerns. We are interested in the possibility of publishing your paper in Nature Methods, but would like to consider your response to these concerns before we reach a final decision on publication.

We therefore invite you to revise your manuscript to address these concerns. In particular, please provide all technical descriptions of the microscope including a parts list, with sufficient detail that other users may be able to build it themselves.

[Redacted]

This URL links to your confidential home page and associated information about manuscripts you may have submitted, or that you are reviewing for us. If you wish to forward this email to co-authors, please delete the link to your homepage.

We hope to receive your revised paper within eight weeks. If you cannot send it within this time, please let us know. In this event, we will still be happy to reconsider your paper at a later date so long as nothing similar has been accepted for publication at Nature Methods or published elsewhere.

--

OPEN SCIENCE REQUIREMENTS

REPORTING SUMMARY AND EDITORIAL POLICY CHECKLISTS

DATA AVAILABILITY

All novel DNA and RNA sequencing data, protein sequences, genetic polymorphisms, linked genotype and phenotype data, gene expression data, macromolecular structures, and proteomics data must be deposited in a publicly accessible database, and accession codes and associated hyperlinks must be provided in the "Data Availability" section.

Please include a "Data availability" subsection in the Online Methods. This section should inform

readers about the availability of the data used to support the conclusions of your study, including accession codes to public repositories, references to source data that may be published alongside the paper, unique identifiers such as URLs to data repository entries, or data set DOIs, and any other statement about data availability. At a minimum, you should include the following statement: "The data that support the findings of this study are available from the corresponding author upon request", describing which data is available upon request and mentioning any restrictions on availability. If DOIs are provided, please include these in the Reference list (authors, title, publisher (repository name), identifier, year). For more guidance on how to write this section please see: <http://www.nature.com/authors/policies/data/data-availability-statements-data-citations.pdf>

CODE AVAILABILITY

Please include a "Code Availability" subsection in the Online Methods which details how your custom code is made available. Only in rare cases (where code is not central to the main conclusions of the paper) is the statement "available upon request" allowed (and reasons should be specified).

For more information on our code sharing policy and requirements, please see: <https://www.nature.com/nature-research/editorial-policies/reporting-standards#availability-of-computer-code>

MATERIALS AVAILABILITY

SUPPLEMENTARY PROTOCOL

To help facilitate reproducibility and uptake of your method, we ask you to prepare a step-by-step Supplementary Protocol for the method described in this paper. We [encourage authors to share their step-by-step experimental protocols](https://www.nature.com/nature-research/editorial-policies/reporting-standards#protocols) on a protocol sharing platform of their choice and report the protocol DOI in the reference list. Nature Portfolio 's Protocol Exchange is a free-to-use and open resource for protocols; protocols deposited in Protocol Exchange are citable and can be linked from the published article. More details can found at www.nature.com/protocolexchange/about.

ORCID

Sincerely,
Madhura

Madhura Mukhopadhyay, PhD
Senior Editor
Nature Methods

Reviewers' Comments:

Reviewer #1:

Remarks to the Author:

The authors present a new light-sheet microscope design that is uniquely capable of 1) accommodating multiple specimens (open-top) with 2) dual view imaging (opposing objectives). They spotlight this new system by imaging larger specimens, including organoids, gastruloids, and Hydra, with quantitative metrics and comparisons to alternative light-sheet systems.

Overall, this work is novel and significant to the field, presenting yet another powerful light-sheet geometry. The data and methodology used throughout the manuscript substantiate the claims and conclusions. The manuscript is also clearly and concisely written.

I have one major suggestion to improve the manuscript. From the current manuscript, the exact specifications of this new microscope are unclear. It would be helpful to include these, and where possible or relevant, draw comparisons in Extended Data Figure 3.

What is the resolution of the system, is it limited by the detection lens NA or pixel sampling? What is the field of view of the system? What are the properties of the light sheet, is it Gaussian and if so what is the illumination NA? Does this cover the entire field of view, or is some tiling approach used? Is the intensity profile across the width of the light sheet Gaussian, or is it digitally scanned? What is the exposure time for these experiments, etc.? What volumetric rate is achievable?

Adding these details to the manuscript would help readers better understand this microscope in the

context of other light sheet microscopes.

Another minor comment is that the manuscript emphasizes the multiple sample aspect. This is certainly possible given the presented design. However, perhaps I missed this, none of the presented imaging results explicitly demonstrate imaging multiple samples simultaneously? Is that correct? If so, would it be more convincing to include an example panel of multiple specimens imaged at the same time?

Reviewer #2:

Remarks to the Author:

The manuscript by Moos et al. presents the development of an open-top dual detection light sheet microscope designed to accommodate larger specimens for live cell imaging. In this work, the authors demonstrate the capacity of the microscope to perform live imaging of different samples, including Hydra and gastruloids. The manuscript shows many time-lapse recordings to demonstrate the power of dual view detection and the advantages of using their Multi-position multi-scale mounting chamber. It's worth noting that this microscope design has been commercialized by Viventis Microscopy, with a significant portion of the contributing authors associated with the company. The manuscript is articulated coherently, accompanied by well-detailed figures. Furthermore, the supplementary videos are visually compelling, offering readers an immersive understanding of time-lapse recordings and three-dimensional perspectives.

However, while I acknowledge the microscope's notable potential in live imaging of 3D structures like organoids and smaller organisms, the manuscript leans more towards demonstrating its applications rather than elucidating the fundamental technical innovations behind the microscope.

MAJOR

I question the uniqueness and novelty of the dual-illumination, dual-view microscope design presented in this work. In the first paragraph of the main text (lines 27-46), the authors describe limitations of other microscope configurations for imaging "large" tissue volumes. However, the citations they rely on seem outdated, especially given that there are recent publications that employ the "dual illumination and dual view" concept, for example, Yang Bin et al., Nature Methods volume 19, pages 461–469 (2022). The authors must do a more thorough review of the recent literature and reframe their presentation to more accurately assert the uniqueness of their contribution.

In the third paragraph (lines 58-81), the narrative delves into the development of the microscope. However, the description will benefit from a more in-depth examination of its technical advancements. A particular statement reads, "To obtain two opposing light sheets illuminating the sample at a largest possible angle we used ...". Explain why it is important to illuminate the sample at the largest possible angle. Moreover, there's just a brief mention of the environment-controlled chamber: "The objective area is temperature controlled and the sample is enclosed in a compartment with controlled CO₂ concentration". This chamber plays a pivotal role, especially for extended time-lapse recordings. Maintaining a stable environment prevents potential distortions and ensures the sample remains in its natural state throughout the recording process. A more thorough description is needed.

"The study showcases the potential of their microscope by imaging an array of samples. Yet, a

significant majority of these experimental observations hinge on $n=1$ experiments. An exception to this is noticeable in Figure 2k,m and Extended Data Fig 2l,m where 3 distinct gastruloids form the basis of the observation. This raises pertinent questions: Do all these observations stem from a singular experiment? Were these 3 gastruloids captured simultaneously within a shared field-of-view, or are they the results of separate observational sessions? To solidify the findings and enhance the study's credibility, the inclusion of quantitative data, complemented by statistical analyses, is crucial. A broader sample base is essential to confirm the reliability and reproducibility both of the microscope's performance and the stability of the environment-controlled chamber. Moreover, the manuscript introduces experiments that utilize the cell cycle probe FUCCI. FUCCI serves as an exemplary probe to monitor cellular activities over extended durations and determine their well-being. A statistical analysis of the FUCCI results would bolster the manuscript."

MINOR

Figure 2a-c, Figure 2c has dotted yellow lines that (I guess) refer to the dotted line in Fig 2a. However, in the legend of Fig 2c it says, "organoid shown in b)".

The code for merging duplicate images is one of the core developments of this work which should be available online.

Line 332-333, correct the sentence "The presented light sheet microscope consists of two illumination, two detection and a transmitted light beam paths."

Legend Figure 2k, correct "...minor axis 42h, 66h and 90 h after ..."

Supplementary Videos (files) are not labeled, e.g., what Supplementary Video appears in 63546_1_video_663298_s0tryb.mp4.

Reviewer #3:

Remarks to the Author:

Summary:

The manuscript discussed the challenges of visualizing the dynamics of individual cells in complex tissues, especially given the vast spatial and temporal scales involved in multicellular systems. Light sheet microscopy, particularly multi-view or SimView light-sheet microscopy, offers some solutions but has limitations. The authors introduced an advanced light sheet microscope that combines multi-view imaging with an open top geometry and a multi-well sample holder. This allowed for long-term, multi-position 3D live imaging of large multicellular systems. The novel open top, dual view, and dual illumination light sheet microscope combines the multiple imaging techniques, allowing for extended 3D live imaging of multicellular systems. This state-of-the-art microscope is characterized by its low phototoxicity, optical sectioning, and improved image quality through multi-view techniques. With it, the authors showed clarity in live imaging of various biological models, from murine organoids to the Hydra. They revealed processes and dynamics previously unseen which reached sizes of up to 550 μm and recordings for up to 12 days. The unique dual detection mechanism of this microscope enhances image quality, ensuring optimal visualization even in the depths of biological samples. Notably, this advancement has permitted time-lapse imaging of the Hydra regeneration process at an unmatched

time resolution, surpassing prior capabilities. Its capacity for high-quality single-cell resolution data was further highlighted through live imaging of intestinal organoids containing the FUCCI2 cell cycle reporter. The chamber design supports post-imaging sample fixation, enabling the fusion of dynamic fluorescent reporter data with functional readouts from antibody staining. Even after removal for immunofluorescent staining, the sample showed minimal movement, enabling effective 3D registration when re-imaged. The Paneth cell marker, Lysozyme (Lys), and the secretory cell marker, Dll1, were employed, which aided in the identification, annotation, and tracking of specific cells. Observations on Paneth cell maturation and their cell cycle arrest in G0/G1 were made, noting the predictive nature of the initial Paneth cell positions regarding the eventual organoid crypt location. Dual illumination and detection imaging techniques enabled observations at depths that would be challenging with single detection. Information from immunofluorescence helped contrast different cell types, revealing that some enterocytes and Paneth cells were already differentiated before imaging. Meanwhile, some cells, particularly in the crypt, were observed during division, with an average cell cycle length of 34.5 hours. It was observed that Paneth cells appear earlier than other cell types in the crypt and villus, suggesting varying cell cycle durations and emergence orders among cell types. Furthermore, this cutting-edge apparatus can simultaneously image under four distinct conditions. The article also demonstrated the ability of their microscope system to provide high-quality single-cell resolution data, especially in dense structures like gastruloids.

The innovations detailed in the article regarding light sheet microscopy are truly noteworthy. Achieving visualization of individual cellular dynamics within intricate tissues over prolonged durations is pivotal for deeper insights into diverse biological phenomena. The introduction of a microscope that melds multi-view imaging with an open-top design and a versatile multi-well sample holder marks a substantial stride in this realm. The article's meticulous experiments and demonstrative analyses proficiently highlight the microscope's potential. Overall, the manuscript is well written and shows the feasibility for the biological community especially in organoid field within the size of 500 μm . The followings are my suggestions before its publication.

Questions:

1. Overall, the images in the figure should be made again to highlight the single cell resolution from the large specimens. It is just hard to appreciate the resolution with the provided figures. Mostly, they are MIP images, no 3D rendering. The reviewer thinks the visualization of the data should be improved to convince the readers. The movies do the good job for the temporal resolution and spatial resolution. Just for the figure, especially for Figure 1, there is some room to point out the spatial resolution or image quality. Also a 3D rendering for the microscope design is needed for easily understanding the geometry for the objectives and chamber system. For the data fusion, the authors should emphasize on the cellular level matching since the single cell resolution is present. It is hard to see from the current figures. For an example, Figure 1 h and j, the middle lower part of the image is ambiguous even after fused.
2. Since this is a technical design for the improvement of the imaging quality over the depth, the reviewer would like to see more lightsheet characterization and details. There is not enough information for lightsheet such as lightsheet length, FWHM, excitation NA, and so on. Is the lightsheet length is a variable parameter? How to change it within the proposed setup? The merit of lightsheet is its optical sectioning, we need the information to see the improvement of the sectioning capacity. The two opposing detection objectives are NA=0.8 WD=3.5 mm, it will be nice to show the detection PSF and overall PSF comparison from the system. Otherwise, the improvement is not easily observed. The

reviewer also holds the concern about the improvement is mostly coming from the inherent sectional capability of the combination of two detection objectives themselves, Nikon 16x NA0.8.

3. The steady lightsheet beam image from dye solution could be provide especially with and without the use of a custom designed correction triplet lens which can compensate for chromatic and spherical aberrations. The measurement the full width at half maximum of the beam to show how well the beam has been improved.

4. In the manuscript, there is little information for the combinations of two lightsheets and two detection objectives, how do they make confocal plane happen? And how the 3D scanning performed by sample scanning. There should be more descriptions or figures. Where the two lightsheet positioned with respect to the detection objective while scanning the samples.

5. The beautiful thing in the present work is about the customized chambers made from FEP foils by thermoforming process to benefit the microscope's efficiency. It would be nice to show some pictures for the real chambers for the systems so that the readers could appreciate how the fabrication is done.

6. In Line 61, "the system is also compatible with Nikon 25X, NA 1.1", this raises the reviewer's interest about what's the performance of this configuration. Since people are shooting for high spatial resolution, of course, high NA detection objective is the solution. However, I didn't see the data obtained in such a high NA configuration, if there is please specify it. If not, is there any constraints for such a configuration? The WD= 2mm for Nikon 25x NA1.1, there should be a room for the micro-FEP chambers. If the reviewer is a user, I would choose high NA one for pursuing high resolution.

7. In Line 76, about the degradation of image quality, the reviewer suggested that measured PSFs of fluorescent beads mixed with Matrigel at the specified concentrations in 3D space over 500 um depth is needed.

8. In Line 93, a dual detection enables the better imaging quality. The recording of the lightsheet images are acquired alternatively in this setup. Is it possible to do the simultaneous detection for the two collection objectives? What's the pros and cons for the proposed setup compared to simultaneous one? The reviewer could imagine that the out-of-focus background will be take an important role while doing the simultaneous detection or this could be removed by lightsheet position with respect to the separate detection objective.

9. In line 103, the performance of the Hydra imaging of the therein lightsheet microscope in reference [20] published in 2018 (already 5 years ago), not sure this is good comparison, maybe shows the exposure time 10ms vs 30 ms for the same fusion proteins? Instead, the reviewer would be more interested in the spatial resolution comparison, the reported publication used 4x objective but still cover the entire Hydra. It would be nice to add the spatial resolution comparison to strengthen the present lightsheet microscope.

10. In line 113, the configuration of Nikon 10x NA0.3 water objective coupled with Nikon 25x NA1.1, the reviewer thinks that it is not feasible due to the physical profiles of these two objectives, unless the lightsheet is very long, but this won't merit the optical sectioning capability. Please check the configuration.

11. Again the beauty of the work is FEP chambers for the parallel imaging of the multi-well chambers. The reviewer is interested in the cell viability, the film is permeable and does the polymer birefringence affect the imaging quality? More descriptions about such a chamber will benefit the people who would like to follow the protocol.

12. Figure 2 gives a good example for the spatial resolution and temporal resolution, also the low phototoxicity of the proposed technique.

13. In line 350, a 10X water immersion objective with an NA of 0.2 (T Plan EPI SLWD 10X, Nikon) and a glass", The T Plan EPI SLWD 10X should be a dry objective, please confirm this.

14. In line 351: "The illumination beam reaches the sample at an angle of 30° with the horizontal axis crossing an air glass and glass water interfaces." The glass window is in front of the illumination

objective, which is orthogonal to the illumination light path?

15. In terms of the code provided by the authors, there are some issues shown here, please check these,

1. The link in github's installation instructions missing a "s"

```
pip install scipy==1.10.0 seaborn==0.12.2 pandas==1.5.3 tifffile==2021.7.2 cikit-
image==0.20.0.dev0 numpy==1.23.5 matplotlib==3.7.0 dipy==1.7.0
```

should be

```
pip install scipy==1.10.0 seaborn==0.12.2 pandas==1.5.3 tifffile==2021.7.2 scikit-
image==0.20.0.dev0 numpy==1.23.5 matplotlib==3.7.0 dipy==1.7.0
```

2. Multiple modules are required during execute the codes, including: "imagecodes" for DCT_plotter.py, "lazy_loader" for ExtractFeature.py and "cellpose" for RunCellpose.py.

3. 1_Organizing_LS2Data.py in The LS2_pipeline can not run and show some path correlated error. The LS2_pipeline shows some path correlated error due to different environments / OS being used. We are testing the code with anaconda and pyCharm under Ms Windows and need modified code correlated with path on all three programs in the pipeline.

For 3_fuse_views_with_sigmoidal.py, we have to modified the code as following:

Insert a line at 252: if `__name__ == "__main__"`:

and change last line (276) from `f.results()` to `f.result()`

Overall, the authors present a new type of open top lightsheet microscope with novel arrangement at illumination arm that allow more space for multiple sample fit into dual illumination and dual detection configuration. Compared with the author's previous setup features with two illuminations but only one detection, the new system extends usable depth two times by dual view fusion from opposition detection. With the FEP chamber design, the setup enables the fast and high image quality for 3D sample less than 500 um. The reviewer thinks the present setup will benefit the community, especially for the organoid field.

Author Rebuttal, first revision:

Reviewer #1:

Remarks to the Author:

The authors present a new light-sheet microscope design that is uniquely capable of 1) accommodating multiple specimens (open-top) with 2) dual view imaging (opposing objectives). They spotlight this new system by imaging larger specimens, including organoids, gastruloids, and Hydra, with quantitative metrics and comparisons to alternative light-sheet systems.

Overall, this work is novel and significant to the field, presenting yet another powerful light-sheet geometry. The data and methodology used throughout the manuscript substantiate the claims and conclusions. The manuscript is also clearly and concisely written.

I have one major suggestion to improve the manuscript. From the current manuscript, the exact specifications of this new microscope are unclear. It would be helpful to include these, and where possible or relevant, draw comparisons in Extended Data Figure 3.

What is the resolution of the system, is it limited by the detection lens NA or pixel sampling? What is the field of view of the system? What are the properties of the light sheet, is it Gaussian and if so what is the illumination NA? Does this cover the entire field of view, or is some tiling approach used? Is the intensity profile across the width of the light sheet Gaussian, or is it digitally scanned? What is the exposure time for these experiments, etc.? What volumetric rate is achievable?

Adding these details to the manuscript would help readers better understand this microscope in the context of other light sheet microscopes.

We thank the reviewer for the positive assessment and constructive feedback about our manuscript. We do agree with the reviewer that additional specifications of the presented microscope are important and thus included all specifications mentioned above in a supplementary table (Supplementary Table 1). In addition, we now included a comparison of the microscope parameters for the approaches which we already compared in Extended Data Figure 6.p. We further also extended the Methods section of the manuscript to increase its comprehensiveness.

Although we now also provide this information in the manuscript, for convenience, we also directly summarize the requested specifications here. Our microscope is limited by the pixel sampling. The pixel size is 0.406 μm , which is larger than the theoretical FWHM of the PSF for the detection NA which is (0.35 μm for 530 nm emission). The cameras capture a FOV of 935 μm . We use a scanned gaussian beam with a FWHM of 3.5 μm (calculated effective illumination NA of 0.06). We did not use a tiling approach for the presented acquisitions. The beam provides an acceptable image quality across most of the field of view. An image of the beam is now shown in Extended Data Figure 2 together with quantification of its FWHM as well as PSF measurements in Matrigel. The overview of the exposure time used for the experiments is now presented in Supplementary Table 2. The volumetric rate easily achievable in a realistic experiment on a real sample is illustrated in the experiment addressing the next review-comment below. The theoretical limit is given by the speed of sample/stage movement and minimal exposure time is 20 planes per second giving for a volume of 200 planes a time interval of around 10s.

Another minor comment is that the manuscript emphasis the multiple sample aspect. This is certainly possible given the presented design. However, perhaps I missed this, none of the presented imaging results explicitly demonstrate imaging multiple samples simultaneously? Is that correct? If so, would it be more convincing to include an example panel of multiple specimens imaged at the same time?

We thank the reviewer for the suggestion. We now highlight Supplementary Video 11, which shows the acquisition of 25 organoids in parallel, better. These organoids were located across all 4 wells of the multi-well sample holder. 201 planes spanning a stack size of 400 μm were imaged at each position with a time

interval of 10 minutes for almost one day. Every position imaged is shown in a separate panel of the video. We believe that this experiment demonstrates the multi sample capability of the system.

Reviewer #2:

Remarks to the Author:

The manuscript by Moos et al. presents the development of an open-top dual detection light sheet microscope designed to accommodate larger specimens for live cell imaging. In this work, the authors demonstrate the capacity of the microscope to perform live imaging of different samples, including Hydra and gastruloids. The manuscript shows many time-lapse recordings to demonstrate the power of dual view detection and the advantages of using their Multi-position multi-scale mounting chamber. It's worth noting that this microscope design has been commercialized by Viventis Microscopy, with a significant portion of the contributing authors associated with the company. The manuscript is articulated coherently, accompanied by well-detailed figures. Furthermore, the supplementary videos are visually compelling, offering readers an immersive understanding of time-lapse recordings and three-dimensional perspectives.

However, while I acknowledge the microscope's notable potential in live imaging of 3D structures like organoids and smaller organisms, the manuscript leans more towards demonstrating its applications rather than elucidating the fundamental technical innovations behind the microscope.

MAJOR

I question the uniqueness and novelty of the dual-illumination, dual-view microscope design presented in this work. In the first paragraph of the main text (lines 27-46), the authors describe limitations of other microscope configurations for imaging "large" tissue volumes. However, the citations they rely on seem outdated, especially given that there are recent publications that employ the "dual illumination and dual view" concept, for example, Yang Bin et al., Nature Methods volume 19, pages 461–469 (2022). The authors must do a more thorough review of the recent literature and reframe their presentation to more accurately assert the uniqueness of their contribution.

We appreciate the reviewers feedback. In the current version of the manuscript, we added a more extensive overview on the recent literature and different light sheet technologies, highlighting the novelty and uniqueness of our work. Further, in our discussion part and Extended Data Fig 6 we also compare our microscope to more recent work. To address the reviewer comment more specifically: In the current version of the manuscript we cite and discuss the publication by Yang Bin et al. This paper presents a single objective light sheet microscope which is a solution sharing the same advantages as the open top light sheet and is in addition compatible with multi well glass bottom plates. Yang Bin et al. are able to illuminate the sample from two sides and for each illumination they also collect a single view. However, because both views are acquired by the same detection objective the fluorescence signal is only collected from one side, which is limiting when imaging large samples and structures further away from the

detection objective. This concept is now illustrated in Extended Data Fig 6 in panel c and in the table below, where we have columns “dual view” and “opposing detection views” to report the differences between the two configurations. We have also added a more detailed explanation of the concept developed by Yang Bin et al. to the introduction of the current version of the manuscript. We hope that these changes to the manuscript address the concern of the reviewer.

In the third paragraph (lines 58-81), the narrative delves into the development of the microscope. However, the description will benefit from a more in-depth examination of its technical advancements. A particular statement reads, “To obtain two opposing light sheets illuminating the sample at a largest possible angle we used ...”. Explain why it is important to illuminate the sample at the largest possible angle.

We thank the reviewer for the comment. In the current version of the manuscript we have now added an explanation for the reasons underling the innovation of this microscope in the main text. In summary: The largest possible angle is needed to minimize striping artifacts and to illuminate the sample uniformly. Indeed, if light enters the sample from one direction it is absorbed, scattered and bend. The intensity of the illumination decays with the distance traveled in the sample and stripes appear due to absorbing objects. To minimize this effect, the sample can be illuminated from an additional direction, ideally directly opposing or if this is not possible (as in our case opposing illumination would prevent multi positioning) it should be done as close as possible to directly opposing, which is at the largest possible angle.

The impact of illuminating with a light sheet at 120 degrees compared to 180 degrees (directly opposing) will be small and largely outweighed by the benefit of two views and multi positioning capabilities. The Extended Data Figure 4 in which we made a comparison of the image quality using different light sheet microscopes compares the same sample imaged with two directly opposing illuminations (panel e, Serra et al, but with one view only) and illuminations at 120 degrees (panel f). As seen from the left half of each panel, the sample is illuminated in both cases homogenously without a noticeable difference.

Moreover, there's just a brief mention of the environment-controlled chamber: “The objective area is temperature controlled and the sample is enclosed in a compartment with controlled CO₂ concentration”. This chamber plays a pivotal role, especially for extended time-lapse recordings. Maintaining a stable environment prevents potential distortions and ensures the sample remains in its natural state throughout the recording process. A more thorough description is needed.

We thank the reviewer for this comment. We agree that the environmental control during the image acquisition is a crucial aspect. To describe this aspect more extensively, we have added a detailed description of the system controlling the environment, in particular the type of CO₂ and temperature sensor used and where they are placed in the microscope.

"The study showcases the potential of their microscope by imaging an array of samples. Yet, a significant majority of these experimental observations hinge on n=1 experiments. An exception to this is noticeable in Figure 2k,m and Extended Data Fig 2l,m where 3 distinct gastruloids form the basis of the observation.

This raises pertinent questions: Do all these observations stem from a singular experiment? Were these 3 gastruloids captured simultaneously within a shared field-of-view, or are they the results of separate observational sessions? To solidify the findings and enhance the study's credibility, the inclusion of quantitative data, complemented by statistical analyses, is crucial. A broader sample base is essential to confirm the reliability and reproducibility both of the microscope's performance and the stability of the environment-controlled chamber. Moreover, the manuscript introduces experiments that utilize the cell cycle probe FUCCI. FUCCI serves as an exemplary probe to monitor cellular activities over extended durations and determine their well-being. A statistical analysis of the FUCCI results would bolster the manuscript."

We agree with the reviewer that one key feature of this microscope is the multi-positional imaging with high sample throughput and therefore, we should highlight it more prominently. We thus increased the sample numbers for quantifications presented in our work. We repeated the experiments involving gastruloids two more times. For each 5.5h time window (42h, 66h, 90h) we analyzed 3 gastruloids. Therefore, 9 gastruloids per time window and 27 gastruloids in total are included in the analysis. We segmented a total of 14356 cells for the 42h time window, 19149 cells for 66h and 21495 cells for the 90h time window. Cell tracking was performed for 62, 56 and 42 cells for the corresponding imaging windows starting 42h, 66h and 90h after cell seeding respectively. Each gastruloid was imaged in an individual field-of-view and several gastruloids were imaged in parallel during one experiment.

Previously our analysis suggested a partial reversion of cell elongation comparing gastruloids imaged for the 60h and 90h imaging window. With increased statistics and sample numbers we see a unidirectional trend without a partial reversion and corrected the manuscript. We determined a median value for the ratio of major/minor axis of 1.89, 2.13 and 2.14 for the 42h, 60h, and 90h time window (Figure 2 k in the manuscript).

The results we obtained for the velocity, track length and mean square displacement of the cells are completely in line with our previous results (Fig 2 m,o,p in the manuscript).

Finally, to increase statistics for our results analyzing intestinal organoids expressing the FUCCI2 cell cycle reporter, we expanded the number of tracked cells by tracking an additional organoid and the results are presented in Extended Data Fig 5 g-l. We additionally identified 4 cells which are positive for hCdt1, Lysozyme and Dll1 at the last time point of the acquisition, 7 hCdt1 positive cells in the villus region and 9 hCdt1 positive cells in the crypt. Including these additionally tracked cells, in the current version of the manuscript we have an average cell cycle length of 21.9 hours at the time of fixation for hCdt1 single positive cells in the crypt.

MINOR

Figure2a-c, Figure 2c has dotted yellow lines that (I guess) refer to the dotted line in Fig2a. However, in the legend of Fig2c it says, "organoid shown in b)".

The code for merging duplicate images is one of the core developments of this work which should be available online.

Line 332-333, correct the sentence “The presented light sheet microscope consists of two illumination, two detection and a transmitted light beam paths.”

Legend Figure 2k, correct “...minor axis 42h, 66h and 90 h after ...”

Supplementary Videos (files) are not labeled, e.g., what Supplementary Video appears in 63546_1_video_663298_s0tryb.mp4.

We thank the reviewer for the detailed comments and addressed all the points in the current version of the manuscript. We also added the code to merge the duplicate views in the public available Github repository [fmi-basel/gliberal-lightsheet-2023 \(github.com\)](https://github.com/fmi-basel/gliberal-lightsheet-2023).

Reviewer #3:

Remarks to the Author:

Summary:

The manuscript discussed the challenges of visualizing the dynamics of individual cells in complex tissues, especially given the vast spatial and temporal scales involved in multicellular systems. Light sheet microscopy, particularly multi-view or SimView light-sheet microscopy, offers some solutions but has limitations. The authors introduced an advanced light sheet microscope that combines multi-view imaging with an open top geometry and a multi-well sample holder. This allowed for long-term, multi-position 3D live imaging of large multicellular systems. The novel open top, dual view, and dual illumination light sheet microscope combines the multiple imaging techniques, allowing for extended 3D live imaging of multicellular systems. This state-of-the-art microscope is characterized by its low phototoxicity, optical sectioning, and improved image quality through multi-view techniques. With it, the authors showed clarity in live imaging of various biological models, from murine organoids to the Hydra. They revealed processes and dynamics previously unseen which reached sizes of up to 550 μm and recordings for up to 12 days. The unique dual detection mechanism of this microscope enhances image quality, ensuring optimal visualization even in the depths of biological samples. Notably, this advancement has permitted time-lapse imaging of the Hydra regeneration process at an unmatched time resolution, surpassing prior capabilities. Its capacity for high-quality single-cell resolution data was further highlighted through live imaging of intestinal organoids containing the FUCCI2 cell cycle reporter. The chamber design supports post-imaging sample fixation, enabling the fusion of dynamic fluorescent reporter data with functional readouts from antibody staining. Even after removal for immunofluorescent staining, the sample showed minimal movement, enabling effective 3D registration when re-imaged. The Paneth cell marker, Lysozyme (Lys), and the secretory cell marker, Dll1, were employed, which aided in the identification, annotation, and tracking of specific cells. Observations on Paneth cell maturation and their cell cycle arrest in G0/G1 were made, noting the predictive nature of the initial Paneth cell positions regarding the eventual

organoid crypt location. Dual illumination and detection imaging techniques enabled observations at depths that would be challenging with single detection. Information from immunofluorescence helped contrast different cell types, revealing that some enterocytes and Paneth cells were already differentiated before imaging. Meanwhile, some cells, particularly in the crypt, were observed during division, with an average cell cycle length of 34.5 hours. It was observed that Paneth cells appear earlier than other cell types in the crypt and villus, suggesting varying cell cycle durations and emergence orders among cell types. Furthermore, this cutting-edge apparatus can simultaneously image under four distinct conditions. The article also demonstrated the ability of their microscope system to provide high-quality single-cell resolution data, especially in dense structures like gastruloids.

The innovations detailed in the article regarding light sheet microscopy are truly noteworthy. Achieving visualization of individual cellular dynamics within intricate tissues over prolonged durations is pivotal for deeper insights into diverse biological phenomena. The introduction of a microscope that melds multi-view imaging with an open-top design and a versatile multi-well sample holder marks a substantial stride in this realm. The article's meticulous experiments and demonstrative analyses proficiently highlight the microscope's potential. Overall, the manuscript is well written and shows the feasibility for the biological community especially in organoid field within the size of 500 μm . The followings are my suggestions before its publication.

Questions:

1. Overall, the images in the figure should be made again to highlight the single cell resolution from the large specimens. It is just hard to appreciate the resolution with the provided figures. Mostly, they are MIP images, no 3D rendering. The reviewer thinks the visualization of the data should be improved to convince the readers. The movies do the good job for the temporal resolution and spatial resolution. Just for the figure, especially for Figure 1, there is some room to point out the spatial resolution or image quality.

We thank the reviewer for the positive comments. We agree that the spatial resolution given by the detection objective is not illustrated in Figure 1 and that 3D renderings are not shown. In the current version of the manuscript, we have therefore adapted Figure 1, which now shows a 3D rendering of a liver organoid in panel I. It also includes a zoom-in showing individual cells both in the 3D rendering and in a maximum intensity projection. The newly added data nicely showcases the single cell resolution. In the case of the hepatic organoid shown, multinucleated hepatocytes, as also described *in vivo*, can be observed. Furthermore, we have now added a movie of a 3D rendering of the liver organoids (Supplementary Video 6).

Also a 3D rendering for the microscope design is needed for easily understanding the geometry for the objectives and chamber system.

To address this comment, in the current version of the manuscript we have now added 3D schematics of the objective arrangement and the position of the sample holder to Extended Data Figure 1b. We believe this improves the illustration of the geometry of our microscope design.

For the data fusion, the authors should emphasize on the cellular level matching since the single cell resolution is present. It is hard to see from the current figures. For an example, Figure 1 h and j, the middle lower part of the image is ambiguous even after fused.

We understand the concern that during image fusion a potential misalignment between the views may cause a mismatch between cells in the area near the switch or in an area where information from both views is used. To illustrate the correspondence between the two views, we now have added an additional panel (Extended data Figure 5 f) showing a XZ section of the gastruloid. The data was fused without a sigmoidal function and therefore a discrete switching point is present (white arrow). Clearly, there is no visible misalignment around the switch point. Additionally red arrows highlight single cells in the single views and their corresponding location in the opposing view where the cells can be easily identified and are at the same location. We believe this panel illustrates that views are aligned and calls between views are matching in space.

2. Since this is a technical design for the improvement of the imaging quality over the depth, the reviewer would like to see more lightsheet characterization and details. There is not enough information for lightsheet such as lightsheet length, FWHM, excitation NA, and so on. Is the lightsheet length is a variable parameter? How to change it within the proposed setup? The merit of lightsheet is its optical sectioning, we need the information to see the improvement of the sectioning capacity. The two opposing detection objectives are NA=0.8 WD=3.5 mm, it will be nice to show the detection PSF and overall PSF comparison from the system. Otherwise, the improvement is not easily observed. The reviewer also holds the concern about the improvement is mostly coming from the inherent sectional capability of the combination of two detection objectives themselves, Nikon 16x NA0.8.

We appreciate the comment of the reviewer and agree that the above-mentioned aspects have to be mentioned in the manuscript. Therefore, we have provided characterization of the light sheet parameters and the PSF of the microscope which can now be found in Extended Data Figure 2. We obtained a lateral resolution of 0.8 μm and an axial resolution of 2.9 μm . It must be mentioned that our resolution is limited by the pixel spacing of our microscope, which is 0.406 μm . The effective NA of the objectives is now estimated in Extended Data Figure 2 and is 0.06. We also added an additional Table (Supplementary Table 1) listing the technical specifications of our microscope.

The thickness of the light sheet has been fixed for all experiments. It is however possible to change it by mounting a beam expander and filling the back aperture of the illumination objective. However this option was not used for the presented experiments.

Because the two opposing objectives generate images on two separate cameras, the sectioning ability of each objective is identical to a situation where it is used alone. The improvement in sectioning and imaging

ability is achieved due to combining the optimal image quality from both objectives allowing a fused and overall better image quality across the entire sample volume.

3. The steady lightsheet beam image from dye solution could be provide especially with and without the use of a custom designed correction triplet lens which can compensate for chromatic and spherical aberrations. The measurement the full width at half maximum of the beam to show how well the beam has been improved.

In the current version of the manuscript, we have provided the characterization and the image of the beam generating the light sheet and it is now shown in Extended Data Fig 2. The beam waist is $3.5 \mu\text{m}$.

To compare the beam with and without correction triplet we performed a simulation shown in Figure 1. The reviewer will appreciate the significant impact of this correction element (Figure 2).

Figure 1: Spot diagram with correction triplet (left) and without (right). Black circle shows airy diameter.

Figure 2: Back focal length (BFL) for different wavelengths showing the chromatic aberration with correction triplet (left) and without (right). The boxes on the top right highlight the BFL for specific wavelengths.

4. In the manuscript, there is little information for the combinations of two lightsheets and two detection objectives, how do they make confocal plane happen? And how the 3D scanning performed by sample scanning. There should be more descriptions or figures. Where the two lightsheet positioned with respective to the detection objective while scanning the samples.

We thank the reviewer for his comment and apologize if some descriptions were not clear. To address this comment, we have now added a paragraph at the end of the Microscope description in the Methods section explaining that the two light sheets must overlap together with the two focal planes as well as providing an alignment procedure to achieve that.

“To acquire one sample plane, both cameras must be positioned to have a common focal plane and both light sheets must be aligned to be in the focal plane of the cameras. The alignment can be done by the following procedure: First, the illumination beams are moved with a galvanometric mirror to be in the focal plane of one camera. Then, the second camera is moved with micrometer screws to be aligned with the illumination beams and focal plane of the first camera.

The light sheet is generated by scanning of the Gaussian beam within the focal plane. To acquire an image plane, the sample is illuminated within the camera exposure time first from one side and then from the other side. Two cameras acquire the two views simultaneously and the data coming from the two views can be fused after the acquisition to one stack. To generate a 3D stack, the sample is moved and the focal planes and light sheets are kept in a fixed position.”

5. The beautiful thing in the presented work is about the customized chambers made from FEP foils by thermoforming process to benefit the microscope’s efficiency. It would be nice to show some pictures for the real chambers for the systems so that the readers could appreciate how the fabrication is done.

We thank the reviewer for this comment. The chambers are fabricated using thermoforming and the current section “Chamber fabrication” in the Methods describes this process in detail. In the manuscript we also refer to a publication of Hötte et al. (2019) which describes this thermoforming process. To further illustrate the manufacturing of the chambers we have added Extended Data Figure 3 showing the specific fabrication steps and some pictures of the chambers.

6. In Line 61, “the system is also compatible with Nikon 25X, NA 1.1”, this raises the reviewer’s interest about what’s the performance of this configuration. Since people are shooting for high spatial resolution, of course, high NA detection objective is the solution. However, I didn’t see the data obtained in such a high NA configuration, if there is please specify it. If not, is there any constraints for such a configuration? The WD= 2mm for Nikon 25x NA1.1, there should be a room for the micro-FEP chambers. If the reviewer is a user, I would choose high NA one for pursuing high resolution.

We thank the reviewer for highlighting this sentence. This sentence in the manuscript was added to indicate that the published concept would also be compatible with the highest NA water immersion objective (Nikon 25x NA 1.1). The working distances mechanically allow the placement of FEP chambers between two 25X detection objectives. However, the presented microscope is only equipped with the 16X objectives.

7. In Line 76, about the degradation of image quality, the reviewer suggested that measured PSFs of fluorescent beads mixed with Matrigel at the specified concentrations in 3D space over 500 μm depth is needed.

We thank the reviewer for the suggestion. We performed the proposed experiment and in the current version of the manuscript it is shown in Extended Data Figure 2 e - g. specifically we show the PSFs over a distance of 400 μm . Even though the molds to produce the FEP chambers were designed to have a width of 500 μm , imprecisions in the manufacturing of the molds results in an effective width of 400 μm inside the FEP chambers. Within that depth in Matrigel, the PSFs were only slightly affected by aberrations induced by Matrigel, as can be seen from the quantifications of the FWHM lateral and axial of exemplary PSFs within different depths inside the chamber.

8. In Line 93, a dual detection enables the better imaging quality. The recording of the lightsheet images are acquired alternatively in this setup. Is it possible to do the simultaneous detection for the two collection objectives? What’s the pros and cons for the proposed setup compared to simultaneous one? The reviewer could imagine that the out-of-focus background will be take an important role while doing the simultaneous detection or this could be removed by lightsheet position with respect to the separate detection objective.

Dual detection achieves better image quality in areas where the image from one detection objective is better than the image of the other detection objective due to the different paths the emitted light travels inside the scattering sample.

Both views are taken simultaneously. To clarify this point, we have added a paragraph to the Methods section explaining this more extensively. Because the focal planes of the two detection objectives overlap and the light sheet is aligned in the same focal plane there is no advantage in imaging the two views sequentially. Illuminating the sample twice, however, will increase the light dose and phototoxicity.

9. In line 103, the performance of the Hydra imaging of the therein lightsheet microscope in reference [20] published in 2018 (already 5 years ago), not sure this is good comparison, maybe shows the exposure time 10ms vs 30 ms for the same fusion proteins? Instead, the reviewer would be more interested in the spatial resolution comparison, the reported publication used 4x objective but still cover the entire Hydra. It would be nice to add the spatial resolution comparison to strengthen the present lightsheet microscope.

We thank the reviewer for the comment. With our microscope we achieve a higher spatial resolution based on the objective configuration used. We use a 16X objective with an NA of 0.8 in our microscope and the cited publication of Iachetta et al. 2018 uses a 4X objective with an NA of 0.13. Their achievable lateral resolution based on the Rayleigh criteria is 2.4 μm (assuming the emission of GFP). The real measured axial resolution of our microscope is 0.8 μm (Extended Data Fig 2 and Supplementary Table 1). We practically achieve a 3 times higher resolution compared to the theoretically achievable resolution of the published work. The difference in temporal resolution is unlikely resulting from the exposure times. In the Methods section of the paper the authors state that they can acquire one stack in 5 seconds. Hence, acquiring with a higher temporal resolution would have been possible using their system but it's very likely not possible due to phototoxicity. The theoretical light collection efficiency of 0.8 NA objective we used is almost 40 times higher than of 0.13 NA objective and this will have an impact on the illumination light intensity required to collect a comparable amount to emitted light and thus also on the phototoxicity.

10. In line 113, the configuration of Nikon 10x NA 0.3 water objective coupled with Nikon 25x NA1.1, the reviewer thinks that it is not feasible due to the physical profiles of these two objectives, unless the lightsheet is very long, but this won't merit the optical sectioning capability. Please check the configuration.

We apologize if the text and descriptions were not clear. The light sheet microscope mentioned in that paragraph is indeed equipped with this specific objective configuration. This can also be found in the supplementary information in the work of Serra et al., 2019, which is cited in the manuscript. It is true that the mentioned objectives do not theoretically fit together mechanically, that means that the centers of their fields of views can't overlap. It is however possible to offset the focal plane and translate the light sheet. By doing this, we realize the mentioned configuration.

11. Again the beauty of the work is FEP chambers for the parallel imaging of the multi-well chambers. The reviewer is interested in the cell viability, the film is permeable and does the polymer birefringence affect the imaging quality? More descriptions about such a chamber will benefit the people who would like to follow the protocol.

We agree that the FEP chambers are an important part of our microscope. The PSF shown in Extended Data Figure 2 was measured through the FEP membrane. For the numerical aperture and pixel spacing used and for the starting membrane thickness we do not see aberrations which would affect experiments shown in this publication. It would be possible to detect aberrations caused by FEP but a different microscope setup with higher NA objective and finer sampling would be required. We believe this is beyond the scope of this manuscript and we hope the reviewer agrees. FEP membranes have been previously used in light sheet microscopy without affecting sample viability. This is illustrated for example in Strnad et al., 2015 showing imaging of mouse embryos for 3 days, or our work published in Serra et al. 2019 where we show continuous acquisitions of intestinal organoids in a FEP based sample holder for 120 h. Both publications are cited in the our manuscript.

12. Figure 2 gives a good example for the spatial resolution and temporal resolution, also the low phototoxicity of the proposed technique.

We thank the reviewer for the comments.

13. In line 350, a 10X water immersion objective with an NA of 0.2 (T Plan EPI SLWD 10X, Nikon) and a glass", The T Plan EPI SLWD 10X should be a dry objective, please confirm this.

We apologize for the mistake, the objective specification is correct, but it is an air objective. We corrected the text accordingly.

14. In line 351: "The illumination beam reaches the sample at an angle of 30° with the horizontal axis crossing an air glass and glass water interfaces." The glass window is in front of the illumination objective, which is orthogonal to the illumination light path?

The glass windows front and back surfaces are orthogonal to the illumination axis given by the axis of the illumination objective.

15. In terms of the code provided by the authors, there are some issues shown here, please check these,

1. The link in github's installation instructions missing a "s"

```
pip install scipy==1.10.0 seaborn==0.12.2 pandas==1.5.3 tiffio==2021.7.2 cikit-image==0.20.0.dev0
numpy==1.23.5 matplotlib==3.7.0 dipy==1.7.0
```

should be

```
pip install scipy==1.10.0 seaborn==0.12.2 pandas==1.5.3 tiffio==2021.7.2 scikit-image==0.20.0.dev0
numpy==1.23.5 matplotlib==3.7.0 dipy==1.7.0
```

2. Multiple modules are required during execute the codes, including: "imagecodes" for DCT_plotter.py, "lazy_loader" for ExtractFeature.py and "cellpose" for RunCellpose.py.

3. 1_Organizing_LS2Data.py in The LS2_pipeline can not run and show some path correlated error.

The LS2_pipeline shows some path correlated error due to different environments / OS being used.

We are testing the code with anaconda and pyCharm under Ms Windows and need modified code correlated with path on all three programs in the pipeline.

For 3_fuse_views_with_sigmoidal.py, we have to modified the code as following:

Insert a line at 252: `if __name__ == "__main__":`

and change last line (276) from `f.results()` to `f.result()`

We thank the reviewer for the time dedicated to reviewing the code. We addressed all the points and updated our Github repository accordingly.

Overall, the authors present a new type of open top lightsheet microscope with novel arrangement at illumination arm that allow more space for multiple sample fit into dual illumination and dual detection configuration. Compared with the author's previous setup features with two illuminations but only one detection, the new system extends usable depth two times by dual view fusion from opposition detection. With the FEP chamber design, the setup enables the fast and high image quality for 3D sample less than 500 um. The reviewer thinks the present setup will benefit the community, especially for the organoid field.

We thank the reviewer for this positive summary of our work.

Decision Letter, second revision:

Dear Prisca,

Thank you for submitting your revised manuscript "Open top multi sample dual view light sheet microscope for live imaging of large multicellular systems" (NMETH-BC53563B). It has now been seen by the original referees and their comments are below. The reviewers find that the paper has improved in revision, and therefore we'll be happy in principle to publish it in Nature Methods, pending minor revisions to satisfy the referees' final requests (I think you can address just ref 3 pt 1 and discuss the rest. Please do send us a rebuttal as well) and to comply with our editorial and formatting guidelines.

TRANSPARENT PEER REVIEW

Nature Methods offers a transparent peer review option for new original research manuscripts submitted from 17th February 2021. We encourage increased transparency in peer review by publishing the reviewer comments, author rebuttal letters and editorial decision letters if the authors agree. Such peer review material is made available as a supplementary peer review file. **Please state in the cover letter 'I wish to participate in transparent peer review' if you want to opt in, or 'I do not wish to participate in transparent peer review' if you don't.** Failure to state your

preference will result in delays in accepting your manuscript for publication.

ORCID

Happy holidays!

Sincerely,
Madhura

Madhura Mukhopadhyay, PhD
Senior Editor
Nature Methods

Reviewer #1 (Remarks to the Author):

The authors have appropriately addressed all of my initial comments.

Reviewer #2 (Remarks to the Author):

The authors have satisfactorily responded to all my comments and revised the manuscript accordingly

Reviewer #3 (Remarks to the Author):

The reviewer thanks the authors' effort to make the manuscript more complete and better structured. The reviewer still holds some concerns as following.

(1) With the information of Gaussian beam, provided by Extended Data Fig 3, no scale bar here, and the color map which didn't shows the details, only highlight the higher intensity, should change the

colormap such as fire scheme or others. If the NA excitation is 0.06, with the beam thickness of 3.5 μm , the FOV \sim 900 μm cannot be covered. Since the sweet spot probably along the center (maybe \sim 1/3 of the FOV). please also plot the line profiles along the different positions of the illumination beam.

(2) The xz or xy overall psf also have the color map issues, which makes the readers hard to appreciate the beam and microscope performance. One more important thing to the reviewer is that the measurement is done by under-sampling, which each pixel is \sim 0.4 μm , this is way to large compared to the Nyquist sampling (as indicated in the reply letter the theoretical FWHM of the PSF for the detection NA which is 0.35 μm for 530 nm emission, that means each pixel should be \sim $0.35/2=0.175$ μm based on the Nyquist sampling) if possible the measurements should be done by the right condition.

(3) The merit of lightsheet is that the good optical sectioning, however, in the present data, this benefit is hard to take since the lightsheet thickness (3.5 μm) is a lot thicker than the depth of focus of the detection objective with NA=0.8 (less than 1 μm). If that is the case, the reviewer is also interested in the presented data compared to the wide-field imaging with z scanning by NA 0.8 objective.

Author Rebuttal, second revision:

Reviewer #3:

Remarks to the Author:

The reviewer thanks the authors' effort to make the manuscript more complete and better structured. The reviewer still holds some concerns as following.

(1) With the information of Gaussian beam, provided by Extended Data Fig 3, no scale bar here, and the color map which didn't shows the details, only highlight the higher intensity, should change the colormap such as fire scheme or others. If the NA excitation is 0.06, with the beam thickness of 3.5 μm , the FOV \sim 900 μm cannot be covered. Since the sweet spot probably along the center (maybe \sim 1/3 of the FOV). please also plot the line profiles along the different positions of the illumination beam.

We appreciate the comment of the reviewer, and we added the missing scale bars in Extended Data Fig 3 and changed the colormap to fire. Furthermore, we included additional line profiles along different positions of the beam and quantified the beam width. The results for each line profile are visualized in each line plot individual. We understand that such beam diverges on the edges of the 935 μm FOV. Nevertheless, from our experience this illumination provides an acceptable image quality across most of the field of view.

(2) The xz or xy overall psf also have the color map issues, which makes the readers hard to appreciate the beam and microscope performance. One more important thing to the reviewer is that the measurement is done by under-sampling, which each pixel is \sim 0.4 μm , this is way to large compared to the Nyquist sampling (as indicated in the reply letter the theoretical FWHM of the PSF for the detection

NA which is 0.35 μm for 530 nm emission, that means each pixel should be $\sim 0.35/2=0.175$ μm based on the Nyquist sampling) if possible the measurements should be done by the right condition.

We thank the reviewer for the suggestion and changed the colormap of the PSFs to fire. We agree that the PSF measurements were performed by under-sampling. However, our microscope is equipped with a magnification of 16X and with cameras with a pixel size of 6.5 μm resulting in a pixel spacing of 0.406 μm which forces us to measure the PSFs under-sampled.

(3) The merit of lightsheet is that the good optical sectioning, however, in the present data, this benefit is hard to take since the lightsheet thickness (3.5 μm) is a lot thicker than the depth of focus of the detection objective with $\text{NA}=0.8$ (less than 1 μm). If that is the case, the reviewer is also interested in the presented data compared to the wide-field imaging with z scanning by NA 0.8 objective.

We thank the reviewer for the comment. The merit of light sheet microscopy is not only optical sectioning, but also it has reduced phototoxicity and good exclusion of background due to not illuminating the entire sample (in contrast to wide field and confocal microscopy). It also offers high imaging speeds and allows multi-view imaging [1]–[3].

Conventional wide-field microscopy does not provide exclusion of background coming from the out of focus light [4], [5]. The illumination power of wide field microscopy at each depth within the specimen is equal. This means, the integrated intensity of the PSF is constant (Fig 1). As comparison, the integrated intensity of confocal microscopes have their maximum at the focal plane and observe fluorophores in a region close to the focus, highlighting their optical sectioning capabilities. Confocal microscopy achieves optical sectioning by reducing the out-of-focus light. In light sheet microscopy, optical sectioning is achieved by illuminating mostly the in-focus part of the sample having the same effect on the PSF as reducing the out-of-focus light in the case of confocal.

Therefore, even with a light sheet microscope with a beam thickness of 3.5 μm , which is larger than the depth of field (around 1 μm), we achieve a substantial optical sectioning and exclusion of background light in comparison to wide field microscopy.

Figure 1: Integrated intensity of the PSF as function of depth z . It is constant for widefield microscopes and peaks at the focus for confocal and two-photon microscopes [5].

References:

- [1] J. Huisken, J. Swoger, F. Del Bene, J. Wittbrodt, and E. H. K. Stelzer, "Optical Sectioning Deep Inside Live Embryos by Selective Plane Illumination Microscopy," *Science*, vol. 305, no. 5686, Art. no. 5686, Aug. 2004, doi: 10.1126/science.1100035.
- [2] P. J. Keller, A. D. Schmidt, J. Wittbrodt, and E. H. K. Stelzer, "Reconstruction of Zebrafish Early Embryonic Development by Scanned Light Sheet Microscopy," *Science*, vol. 322, no. 5904, pp. 1065–1069, Nov. 2008, doi: 10.1126/science.1162493.
- [3] U. Krzic, S. Gunther, T. E. Saunders, S. J. Streichan, and L. Hufnagel, "Multiview light-sheet microscope for rapid in toto imaging," *Nat. Methods*, vol. 9, no. 7, Art. no. 7, Jul. 2012, doi: 10.1038/nmeth.2064.
- [4] J. Huisken, "Multi-view microscopy and multi-beam manipulation for high-resolution optical imaging," University Freiburg, 2004.
- [5] J. E. N. Jonkman, J. Swoger, H. Kress, A. Rohrbach, and E. H. K. Stelzer, "Resolution in optical microscopy," in *Methods in Enzymology*, vol. 360, Academic Press, 2003, pp. 416–446. doi: 10.1016/S0076-6879(03)60122-9.

Final Decision Letter:

Dear Prisca,

I am pleased to inform you that your Brief Communication, "Open top multi-sample dual-view light-sheet microscope for live imaging of large multicellular systems", has now been accepted for publication in *Nature Methods*. The received and accepted dates will be 5th Sep, 2023 and 18th Jan, 2024. This note is intended to let you know what to expect from us over the next month or so, and to let you know where to address any further questions.

Over the next few weeks, your paper will be copyedited to ensure that it conforms to *Nature Methods* style. Once your paper is typeset, you will receive an email with a link to choose the appropriate publishing options for your paper and our Author Services team will be in touch regarding any additional information that may be required.

Once proofs are generated, they will be sent to you electronically and you will be asked to send a corrected version within 48 hours. It is extremely important that you let us know now whether you will be difficult to contact over the next month. If this is the case, we ask that you send us the contact information (email, phone and fax) of someone who will be able to check the proofs and deal with any last-minute problems.

If, when you receive your proof, you cannot meet the deadline, please inform us at rjsproduction@springernature.com immediately.

Please note that *Nature Methods* is a Transformative Journal (TJ). Authors may publish their research with us through the traditional subscription access route or make their paper immediately open access through payment of an article-processing charge (APC). Authors will not be required to make a final decision about access to their article until it has been accepted. Find out more about Transformative Journals

If you are active on Twitter/X, please e-mail me your and your coauthors' handles so that we may tag you when the paper is published.

To assist our authors in disseminating their research to the broader community, our SharedIt initiative provides you with a unique shareable link that will allow anyone (with or without a subscription) to read the published article. Recipients of the link with a subscription will also be able to download and print the PDF. As soon as your article is published, you will receive an automated email with your shareable link.

Please note that you and your coauthors may order reprints and single copies of the issue containing your article through Springer Nature Limited's reprint website, which is located at <http://www.nature.com/reprints/author-reprints.html>. If there are any questions about reprints please send an email to author-reprints@nature.com and someone will assist you.

Best regards,
Madhura

Madhura Mukhopadhyay, PhD
Senior Editor
Nature Methods